# First national tuberculosis patient cost survey in Lao People's Democratic Republic: Assessment of the financial burden faced by TB-affected households and the comparisons by drug-resistance and HIV status

Phonenaly Chittamany[1], Takuya Yamanaka[2,3,4]*, Sakhone Suthepmany[1], Thepphouthone Sorsavanh[1], Phitsada Siphanthong[1], Jacques Sebert[1], Kerri Viney[2,5,6], Thipphasone Vixaysouk[7], Moeko Nagai[7], Vilath Seevisay[7], Kiyohiko Izumi[7], Fukushi Morishita[8], Nobuyuki Nishikiori[2]

1 National TB Programme, Ministry of Health, Vientiane, Lao PDR, 2 Global Tuberculosis Programme, World Health Organization, Geneva, Switzerland, 3 Department of Global Health and Development, London School of Hygiene and Tropical Medicine, London, United Kingdom, 4 School of Tropical Medicine and Global Health, Nagasaki University, Nagasaki, Japan, 5 Research School of Population Health, Australian National University, Canberra, Australia, 6 Department of Global Public Health, Karolinska Institutet, Stockholm, Sweden, 7 World Health Organization, Country Office, Vientiane, Lao PDR, 8 World Health Organization, Regional Office for the Western Pacific, Manila, Philippines

* Takuya.Yamanaka@lshtm.ac.uk

## Abstract

### Background

Tuberculosis (TB) patients incur large costs for care seeking, diagnosis, and treatment. To understand the magnitude of this financial burden and its main cost drivers, the Lao People's Democratic Republic (PDR) National TB Programme carried out the first national TB patient cost survey in 2018–2019.

### Method

A facility-based cross-sectional survey was conducted based on a nationally representative sample of TB patients from public health facilities across 12 provinces. A total of 848 TB patients including 30 drug resistant (DR)-TB and 123 TB-HIV coinfected patients were interviewed using a standardised questionnaire developed by the World Health Organization. Information on direct medical, direct non-medical and indirect costs, as well as coping mechanisms was collected. We estimated the percentage of TB-affected households facing catastrophic costs, which was defined as total TB-related costs accounting for more than 20% of annual household income.

### Result

The median total cost of TB care was US$ 755 (Interquartile range 351–1,454). The costs were driven by direct non-medical costs (46.6%) and income loss (37.6%). Nutritional

**Data Availability Statement:** Survey data sets contain privacy-sensitive information including participant's individual and household income that formed a core part of the analysis. Even though we remove patient's identifiers such as patient number and name, there is still a possibility that those who are familiar with the project sites and beneficiaries may be able to identify participants and their households. The informed consent signed by all participants explicitly mentioned that only the research team have access to the data set. Due to such ethical and confidentiality restrictions, data sets will be made available only upon request and with permission from World Health Organization and the National Center for Tuberculosis Control, Ministry of Health, Lao PDR. All interested researchers will contact WHO/WPRO ethics review committee (wproethicsreviewcomm@who.int) to request the data access.

**Funding:** The national TB patient cost survey in Lao PDR was financially supported by the Government of the Republic of Korea through Korean Centers for Disease Control & Prevention, and the Government of Japan through Ministry of Health, Labour and Welfare. The additional data collection for drug resistant TB and TB-HIV coinfected patients were funded by WHO/TDR-WPRO small research grant 2018-2019. The funders had no role in study design, data collection and analysis, decision to publish, or preparation of the manuscript.

**Competing interests:** The authors have declared that no competing interests exist.

supplements accounted for 74.7% of direct non-medical costs. Half of the patients used savings, borrowed money or sold household assets to cope with TB. The proportion of unemployment more than doubled from 16.8% to 35.4% during the TB episode, especially among those working in the informal sector. Of all participants, 62.6% of TB-affected households faced catastrophic costs. This proportion was higher among households with DR-TB (86.7%) and TB-HIV coinfected patients (81.1%).

## Conclusion

In Lao PDR, TB patients and their households faced a substantial financial burden due to TB, despite the availability of free TB services in public health facilities. As direct non-medical and indirect costs were major cost drivers, providing free TB services is not enough to ease this financial burden. Expansion of existing social protection schemes to accommodate the needs of TB patients is necessary.

## Introduction

Tuberculosis (TB) is one of the major public health concerns globally [1]. TB patients usually incur a substantial financial cost for care seeking, diagnosis, and treatment [2]. Poverty is already known to be linked to a higher risk of TB infection and disease, delays in diagnosis, and poor treatment adherence, which can result in adverse treatment outcomes and the development of multi-drug resistant (MDR)-TB [2–5]. TB also worsens poverty as TB patients often lose the ability to work, leading to income loss [5–7]. In addition, households affected by TB will often mobilize funds for TB treatment by dissaving, selling assets, or taking out loans, which makes them poorer, and traps them in a cycle of poverty and disease [8–10].

Considering the global burden of TB and the social, human and financial consequences of TB, the World Health Organization (WHO) has developed the End TB Strategy, which outlines the ambitious goal of ending the TB epidemic worldwide by 2030 [11]. Recognizing the need to address the financial burden due to TB, the Strategy promotes assessment of TB patient costs and set a target of zero "catastrophic costs" for TB-affected families, in the context of Universal Health Coverage, in addition to two traditional epidemiological targets (reduced incidence and deaths) [11].

To establish the baseline against which to monitor the progress towards the elimination of catastrophic costs, WHO recommends that a nationally representative survey be carried out. These surveys are prioritised for the 30 high TB burden countries [11]. In 2015, WHO developed a generic protocol and data collection tool to support countries in planning and implementing the national TB patient cost surveys, which were later refined and published as a handbook in 2017 [12]. The WHO-recommended data collection tool is designed to collect a range of cost data including direct medical costs (i.e. costs for medical consultations, examinations, drugs, and hospitalization), direct non-medical costs (i.e. transportation, foods and accommodation), and indirect costs (i.e. loss of income). Based on this information, catastrophic costs attributable to TB can be calculated. "Catastrophic costs due to TB" refers to medical and non-medical out-of-pocket payments and indirect costs exceeding a given threshold (e.g. 20%) of the household's income [11, 12].

The Lao PDR National TB Programme (NTP) has developed their National TB Strategic Plan 2017–2020 which is aligned with the WHO's End TB Strategy and the Regional

Framework for Action on Implementation of the End TB Strategy in the Western Pacific 2016–2020 [11, 13]. One of the key indicators is: "Zero TB affected families facing catastrophic costs due to TB" [14]. It was therefore prerequisite for the NTP to conduct a national TB patient cost survey to establish a baseline of the target and inform policies and strategies. The study objectives were to assess the financial burden of TB from the patient perspective and to estimate the proportion of households experiencing catastrophic costs due to TB, by drug-susceptibility and HIV status. This survey also assessed major cost drivers of patients' expenditure during TB episode and associated risk factors for facing catastrophic costs due to TB.

## Method

### Study setting

Lao PDR is a landlocked country surrounded by China, Viet Nam, Cambodia, and Thailand, and the majority of the country is mountainous and forested [15, 16]. The population size was 6.9 million in 2017, and the majority of the population (60%) live in rural areas [17]. The country is divided into one capital city (Vientiane Capital) and 17 provinces. The capital is located on the banks of Mekong river nearby to the border with Thailand, and the population size was 820,940 in 2015 [18]. Lao PDR has gross domestic product (GDP) growth average at 7.8%, and is a lower middle income country with US$ 14.5 billion in Gross National Income (GNI) and a gross national income per capita of US$ 2,150 per annum in 2016 [19].

Lao PDR has a high burden of TB [1]. The first national TB prevalence survey in Lao PDR, conducted in 2010–2011, revealed that the prevalence of bacteriologically confirmed pulmonary TB was estimated at 237 cases per 100,000 population [20]. According to the WHO's estimates, the TB incidence rate remained high in 2018, at 162 cases per 100,000 population (all forms of TB), while the incidence rate of HIV positive TB and Multidrug-/rifampicin-resistant TB (MDR/RR-TB) were 10 and 2.2 cases per 100,000 respectively [1, 21]. The TB mortality rate in Lao PDR was 33.8 cases per 100,000 population in 2017 [1, 21].

In Lao PDR, approximately 20% of the total population work in the formal sector and are covered by a comprehensive benefit schemes such as State Authority for Social Security (SASS) and Social Security Organization (SSO) under the National Social Security Fund (NSSF). The NSSF includes health insurance, sickness benefits, and unemployment benefits [22]. The tax-based National Health Insurance (NHI) scheme was launched in 2016. The NHI covered 60% of the population as of 2016, and the coverage was expected to reach more than 70% in 2017 [23]. The national health insurance bureau has recently reported that the coverage reached 79.3% in 2019. All health services defined by the Essential Health Service Package (EHSP) at public health facilities in the country are covered by the NHI scheme [24, 25]. Costs for TB diagnosis and treatment and hospitalization for all TB patients are covered by the NTP and NHI who cover the cost of sputum examinations including smear, culture and Xpert MTB/RIF (a rapid molecular test), testing and counselling for HIV, and anti-TB drugs for all TB patients including second-line drugs [25]. Eleven of 25 central and provincial hospitals provide both integrated TB and HIV services, including the provision of both TB treatment and Anti-Retroviral Therapy (ART) [26–28].

### Study design, population, and sample size

Following the WHO recommended study design, we conducted a facility-based cross-sectional survey and extrapolated costs in the patient's current TB treatment phase (i.e. intensive or continuation) to assess the total costs associated with a diagnosis of TB and ongoing TB care [12]. We used a cluster sampling strategy to ensure a nationally representative sample. The primary sampling unit was the Basic Management Unit (BMU) of the NTP; these

are 165 district, provincial, and central hospitals [1, 12]. A total of 25 BMUs was randomly selected by a probability proportional to size (PPS) method applying the TB case notification in 2017 for each BMU [1]. With a design effect of 2.0, an estimated catastrophic cost prevalence of 50%, and a precision level of 5%, the required sample size was 725. We assumed incidence of 50% catastrophic costs in Lao PDR from the previous TB patient cost surveys conducted in the Western Pacific Region (Philippines: 35%, Vietnam: 63%, Mongolia: 70% with an unweighted average of 56%) [29]. An estimated prevalence of 50% provided the most conservative sample size. In addition to this nationally representative sample, we enrolled additional and operationally feasible quotas of 120 TB-HIV co-infected patients and 30 DR-TB (i.e. MDR-TB or RR-TB) patients to assess the difference in the financial burden of TB comparing drug-resistant and drug-susceptible TB patients and for patients with and without TB-HIV co-infection. We enrolled all the patients on DR-TB treatment at the time of interview in the country (total sampling). For TB-HIV patients, assuming estimated proportion of catastrophic costs at 80% with 10% precision and design effect of 2, the sample size was estimated at 122.

All eligible TB patients were those who were currently on TB treatment linked to the NTP (including adults and children) who had received at least 14 days of treatment in either the intensive or continuation phase of TB treatment. The participants were selected randomly from TB log-book at each facility. If a child was recruited, we interviewed the parent of the child participant. We excluded people who were treated in facilities that are unlinked to the NTP (i.e. private facilities which do not report TB cases to the NTP).

## Data collection

We used a standardised questionnaire developed by WHO, adapted this to the country context, and translated into the local language. Twelve interviewers were recruited through an open recruitment process. We conducted a 3-day training for all staff and interviewers involved in the survey in October 2018, and the interview tool was piloted during the training using the Ona online platform [30]. We then undertook face-to-face interviews with randomly selected TB patients at health facilities during their facility visits and entered responses directly into tablets using the ONA online platform [12, 30].

The questionnaire included questions on different types of direct medical costs (e.g. medical consultation, laboratory tests, medications, hospitalization), direct non-medical costs (e.g. transportation, food, accommodation, and nutritional supplements such as vitamin supplements and/or additional foods other than regular diets) and indirect costs (income loss and time lost for care seeking). We also collected demographic and clinical information, information on health care utilization, household assets, coping mechanisms (e.g. dissaving, borrowing, sold assets), and perceived social and financial impacts of a TB diagnosis and care. Each patient was interviewed once and reported on expenditures and time spent for care seeking, coping mechanism, and household assets and income during the current treatment phase (e.g. either the intensive phase or the continuation phase). Total time spent for care seeking was estimated by multiplying time loss for the last visit with frequency of visits per month and the duration of TB treatment. For patients interviewed in the intensive phase, retrospective data on costs and time spent for care seeking before TB diagnosis were also collected, however these questions were not asked of patients in the continuation phase. Data collection was conducted between December 2018 until January 2019, followed by the additional data collection for DR-TB and TB-HIV co-infected patients which was carried out in May and June 2019.

## Data analysis

For continuous data, median, mean, inter-quartile range (IQR), 95% confidence interval (95% CI) were presented, and for categorical data, frequencies were calculated. The median values were used to present information on costs and incomes, due to the skewness of the data. We assessed the difference between DS-TB and DR-TB using Fisher's exact test for categorical data and the Welsh T-test or two-sample Wilcoxon rank-sum test for continuous data. Statistical significance was defined as $p < 0.05$, and statistical analyses and data visualizations were performed using Stata 15.2 (StataCorp 2018) and R4.0.2 (CRAN: Comprehensive R Archive Network at https://cran.r-project.org/). Due to different sampling methods for nationally representative sample and additional sample of DR-TB and TB-HIV coinfected patients, statistical tests were performed only in nationally representative sample. All cost and income data were collected in the local currency (Lao Kip) and then were converted into US$ for analysis at the rate of 8,495 Kip per 1 US$ (oanda.com).

For patient cost data, we calculated median costs with IQR stratified by different cost types (direct medical and direct non-medical costs and indirect costs) from the time of onset of TB symptom until the TB treatment completion. Since we collected only the costs of TB treatment incurred during the treatment phase patients were in at the time of the interview, the costs of the other treatment phase for were extrapolated based on the median costs incurred by other patients in that treatment phase at the time of the interview. For example, to estimate pre-treatment costs and costs during TB intensive phase for patients who were in continuation phase at the time of interview, the median costs of pre-treatment costs and intensive phase were taken from the patients who were in intensive phase at the time of interview. In this calculation for extrapolating costs, costs from DS-TB and DR-TB patients, and patients with and without hospitalizations were considered separately. We estimated income loss in TB patients' households using the income prior to the current TB episode and that at the time of interview (output approach). Monthly self-reported income was used as the primary method for determining household income. However, we also computed predicted annual household income based on a linear regression analysis using household asset information (as we collected information on household assets in the questionnaire). 20% of annual household income was used as the threshold to define catastrophic costs due to TB, consistent with the definition proposed by WHO [12]. Clustering effects associated with sampling method were adjusted in estimating overall proportion of catastrophic costs using the svy command in Stata software. An additional sensitivity analysis was conducted to assess how varying the threshold (to different percentages from 0% to 100%) affected the proportion of catastrophic costs faced by TB-affected households.

Then we carried out a univariate logistic regression analysis to identify demographic and clinical factors associated with facing catastrophic costs due to TB. The sample used in this regression excluded the additional patients recruited who had MDR-TB and TB-HIV co-infection. We included variables in our multivariate logistic regression analysis, if they were significant at the 10% level (P<0.10) in univariate analyses. Clustering effects associated with sampling method were adjusted both in univariate and multivariate analyses using the svy command in Stata software.

## Ethical approval

The survey was approved by the Ethics Review Committee of the WHO Regional Office for the Western Pacific (WHO/WPRO) (Ref: 2018.10.LAO.4.STB) and the Lao PDR National Ethics Committee for Health Research (Ref: 091/NECHR). Written informed consent was obtained from all the survey participants before the commencement of the interview. For participants aged < 15 years of age, we obtained written informed consent from a parent or guardian.

## Results

### Study population

A total of 848 TB patients participated in the survey, and of these 725 patients (717 DS-TB and 8 DR-TB) were enrolled as a nationally representative sample (the national sample) and 123 patients (22 DR-TB and 101 TB-HIV) were enrolled as an additional sample (Table 1). In the national sample, 59.7% were male, and the mean age was 50.4 years. 0.8% were children aged under 15 years old. The proportion of HIV positive patients was 2.6%, while 32.0% had unknown HIV status. The demographic characteristics of the national sample were similar to the data observed and reported in NTP routine surveillance. The proportion of participants with no education or had only attended primary school was 41.2%, and 61.9% had informal paid jobs before their TB diagnosis. The majority of participants (75.9%, N = 550) reported that they were not covered by any health insurance, and only (6.2%, N = 45) were insured by

**Table 1. Socio-demographic characteristics of participants of the national TB patient cost survey by drug resistance and HIV status, Lao PDR, 2018–2019.**

| Characteristics | | Nationally representative sample | | | | | With additional sample | | | |
|---|---|---|---|---|---|---|---|---|---|---|
| | | DS-TB | | DR-TB | | P-value | All | | All DR-TB | | All TB-HIV | |
| | | n = 717 | | n = 8 | | | n = 725 | | n = 30 | | n = 123 | |
| Gender | Female | 291 | (40.6%) | 1 | (12.5%) | 0.153 | 292 | (40.3%) | 14 | (46.7%) | 40 | (32.5%) |
| | Male | 426 | (59.4%) | 7 | (87.5%) | | 433 | (59.7%) | 16 | (53.3%) | 83 | (67.5%) |
| Age | mean (95%CI) | 50.5 | (49.3–51.7) | 41.4 | (27.3–55.4) | 0.125 | 50.4 | (49.2–51.6) | 44.7 | (38.6–50.8) | 33.8 | (32.0–35.7) |
| HIV status | Negative | 466 | (65.0%) | 8 | (100.0%) | 0.103 | 474 | (65.4%) | 27 | (90.0%) | 0 | (0.0%) |
| | Positive | 19 | (2.6%) | 0 | (0.0%) | | 19 | (2.6%) | 2 | (6.7%) | 123 | (100.0%) |
| | Unknown | 232 | (32.4%) | 0 | (0.0%) | | 232 | (32.0%) | 1 | (3.3%) | 0 | (0.0%) |
| Patient's education level | No education | 177 | (24.7%) | 0 | (0.0%) | 0.136 | 177 | (24.4%) | 7 | (23.3%) | 5 | (4.1%) |
| | Primary school | 265 | (37.0%) | 2 | (25.0%) | | 267 | (36.8%) | 11 | (36.7%) | 34 | (27.6%) |
| | Secondary/High school | 181 | (25.2%) | 5 | (62.5%) | | 186 | (25.7%) | 8 | (26.7%) | 63 | (51.2%) |
| | Vocational, University and higher | 73 | (10.2%) | 1 | (12.5%) | | 74 | (10.2%) | 4 | (13.3%) | 14 | (11.4%) |
| | Other | 21 | (2.9%) | 0 | (0.0%) | | 21 | (2.9%) | 0 | (0.0%) | 6 | (4.9%) |
| Occupation (pre-disease) | Not employed | 122 | (17.0%) | 0 | (0.0%) | 0.245 | 122 | (16.8%) | 2 | (6.7%) | 5 | (4.1%) |
| | Employed (formal) | 78 | (10.9%) | 0 | (0.0%) | | 78 | (10.8%) | 3 | (10.0%) | 30 | (24.4%) |
| | Employed (informal) | 443 | (61.8%) | 6 | (75.0%) | | 449 | (61.9%) | 20 | (66.7%) | 77 | (62.6%) |
| | Retired/student/housework | 70 | (9.8%) | 2 | (25.0%) | | 72 | (9.9%) | 5 | (16.7%) | 7 | (5.7%) |
| | Other | 4 | (0.6%) | 0 | (0.0%) | | 4 | (0.6%) | 0 | (0.0%) | 3 | (2.4%) |
| Insurance type | None | 543 | (75.7%) | 7 | (87.5%) | 1.000 | 550 | (75.9%) | 26 | (86.7%) | 99 | (80.5%) |
| | National Health Insurance (NHI) | 32 | (4.5%) | 0 | (0.0%) | | 32 | (4.4%) | 2 | (6.7%) | 6 | (4.9%) |
| | Community-Based Health Insurance (CBHI) | 81 | (11.3%) | 1 | (12.5%) | | 82 | (11.3%) | 1 | (3.3%) | 0 | (0.0%) |
| | Health Equity Fund (HEF) | 6 | (0.8%) | 0 | (0.0%) | | 6 | (0.8%) | 0 | (0.0%) | 3 | (2.4%) |
| | Social Security Organization (SSO) | 3 | (0.4%) | 0 | (0.0%) | | 3 | (0.4%) | 0 | (0.0%) | 4 | (3.3%) |
| | State Authority for Social Security (SASS) | 42 | (5.9%) | 0 | (0.0%) | | 42 | (5.8%) | 1 | (3.3%) | 5 | (4.1%) |
| | Private health insurance | 10 | (1.4%) | 0 | (0.0%) | | 10 | (1.4%) | 0 | (0.0%) | 2 | (1.6%) |
| | Other | 0 | (0.0%) | 0 | (0.0%) | | 0 | (0.0%) | 0 | (0.0%) | 3 | (2.4%) |
| Household size | mean (95%CI) | 5.7 | (5.5–5.9) | 4.5 | (3.7–5.3) | 0.170 | 5.7 | (5.5–5.9) | 4.6 | (3.8–5.4) | 4.4 | (4.1–4.7) |
| Monthly household income | median (IQR), in US$ | 235 | (118–471) | 235 | (141–347) | 0.726 | 235 | (118–471) | 235 | (118–494) | 353 | (200–589) |

IQR: Interquartile range, 95% CI: 95% confidence interval, DS-TB: drug-susceptible TB, DR-TB: drug-resistant TB.

SSO or SASS. The mean household size was 5.7 individuals with a median monthly household income of US$ 235 (IQR: 118–471).

No significant differences were observed in socio-demographic factors when comparing patients with DS-TB from the national sample and patients with DR-TB. In all TB-HIV patients with additional sample (N = 123), the mean age of 33.8 years was considerably lower than the nationally representative sample (the mean age of 50.4 years), and the median monthly household income of US$ 353 was higher than other groups (**Table 1**).

In the national sample, 97.9% were newly diagnosed, and 62.8% were in the continuation phase of TB treatment (**Table 2**). The majority (94.2%) had pulmonary TB, and 68.2% were bacteriologically confirmed. The proportion of patients who were taking TB treatment in a public health centre or a district hospital was 71.0% in the national sample. On the other hand, most of the DR-TB and TB-HIV coinfected patients were receiving TB treatment in provincial or central hospitals (DR-TB: 100% and TB-HIV: 96.7%) with a high incidence of hospitalization during the current treatment phase compared to patients with DS-TB (national sample DR-TB: 75.0% and DS-TB: 11.4% p<0.001. With additional samples, all DR-TB: 90.0% and all TB-HIV: 38.2%).

The median number of weeks reported from the onset of TB symptoms until initiation of TB treatment was 4 weeks, with an average of 1.8 visits to a health facility before a TB diagnosis was made (the maximum number of visits was 19). Just over one third (30.4%, N = 79) of 260 patients who were in the intensive phase initiated their care seeking at private healthcare providers such as traditional healers, private pharmacies, clinics and/or hospitals. The large majority of patients in the national sample (97.7%) were self-administering TB medications, while half of DR-TB patients were receiving directly observed therapy (DOT) while they were hospitalized in the TB ward (national sample, DS-TB: 1.8%, DR-TB: 50.0%, p<0.001. With additional sample, all DR-TB: 56.7%, all TB-HIV: 12.2%). Therefore, visits for drug pick-up was most frequent (15.6 times) in the national sample and for patients with DS-TB, while DR-TB patients had more DOT visits (136.4 times) during TB treatment.

## Time loss for care seeking

The median total time loss for care seeking for TB patients and their caregivers from the onset of TB symptoms until the completion of TB treatment was 95 hours (IQR: 79–139) and 1 hour (IQR: 0–13), respectively, with a significantly larger time loss for DR-TB patients when compared to DS-TB patients in the national sample (patients: DS-TB 95 hours (IQR: 79–139), DR-TB 1,327 hours (IQR: 665–2,517), p = 0.006) (**Table 3**). Time loss associated with the prediagnosis period and hospitalization was also significantly longer among DR-TB patients compared to DS-TB (Pre-diagnosis: DS-TB 9 hours (IQR: 3–44), DR-TB 40 hours (IQR: 32–270), p = 0.050. Hospitalization: DS-TB 76 hours (IQR: 37–158), DR-TB 755 hours (IQR: 301–1,088), p = 0.026). No significant differences were observed in time loss in caregivers in the national sample. With additional sample, TB-HIV patients also reported a considerably longer time spent on hospitalization (patients: 122 hours (IQR: 83–220)).

## Estimated total costs borne by TB patient and their households

The total median cost incurred for TB care was US$ 755 (IQR: US$ 351–1,454); equivalent to 3.2 times the average monthly salary of TB patients in the survey (**Table 4**). Only 9.2% of costs were incurred before a TB diagnosis. The largest cost driver was direct non-medical costs (46.6%), followed by indirect costs (37.6%) and direct medical costs (15.8%) (**Fig 1**). In particular, the costs for special foods and nutritional supplements other than the patients regular diet was high, comprising 34.8% of total costs. Total costs were nearly double for TB-HIV co-

**Table 2. Clinical characteristics of participants of the national TB patients cost survey by drug resistance and HIV status, Lao PDR, 2018–2019.**

| Characteristics | | Nationally representative sample | | | | With additional sample | |
|---|---|---|---|---|---|---|---|
| | | DS-TB | DR-TB | P-value | All | All DR-TB | All TB-HIV |
| | | n = 717 | n = 8 | | n = 725 | n = 30 | n = 123 |
| **Treatment phase** | Intensive phase | 265 (37.0%) | 5 (62.5%) | 0.156 | 270 (37.2%) | 19 (63.3%) | 50 (40.7%) |
| | Continuation phase | 452 (63.0%) | 3 (37.5%) | | 455 (62.8%) | 11 (36.7%) | 73 (59.3%) |
| **Treatment category** | New | 702 (97.9%) | 8 (100.0%) | 1.000 | 710 (97.9%) | 24 (80.0%) | 121 (98.4%) |
| | Relapse | 12 (1.7%) | 0 (0.0%) | | 12 (1.7%) | 2 (6.7%) | 0 (0.0%) |
| | Loss to follow-up or treatment after failure | 2 (0.3%) | 0 (0.0%) | | 2 (0.3%) | 4 (13.3%) | 1 (0.8%) |
| **Type and diagnosis of TB** | Pulmonary TB (Bacteriologically Confirmed) | 488 (68.1%) | 7 (87.5%) | 0.646 | 495 (68.3%) | 28 (93.3%) | 42 (34.1%) |
| | Pulmonary TB (Clinically diagnosed) | 191 (26.6%) | 1 (12.5%) | | 192 (26.5%) | 2 (6.7%) | 67 (54.5%) |
| | Extra pulmonary TB | 38 (5.3%) | 0 (0.0%) | | 38 (5.2%) | 0 (0.0%) | 14 (11.4%) |
| **Planned treatment duration** | 6 months | 715 (97.7%) | 0 (0.0%) | <0.001*** | 715 (98.6%) | 0 (0.0%) | 123 (100.0%) |
| | 9 months | 0 (0.0%) | 8 (100.0%) | | 8 (1.1%) | 30 (100.0%) | 0 (0.0%) |
| | 12 months | 2 (0.3%) | 0 (0.0%) | | 2 (0.3%) | 0 (0.0%) | 0 (0.0%) |
| **Registered facility type** | Public health centre | 133 (18.5%) | 0 (0.0%) | 0.136 | 133 (18.3%) | 0 (0.0%) | 0 (0.0%) |
| | District hospital | 382 (53.3%) | 0 (0.0%) | | 382 (52.7%) | 0 (0.0%) | 4 (3.3%) |
| | Provincial hospital | 134 (18.7%) | 4 (50.0%) | | 138 (19.0%) | 12 (40.0%) | 56 (45.5%) |
| | Central, military or police hospital | 68 (9.5%) | 4 (50.0%) | | 72 (9.9%) | 18 (60.0%) | 63 (51.2%) |
| **From onset of symptom until diagnosis** | median (IQR), in weeks | 4 (2–7) | 8 (2–8) | 0.672 | 4 (2–8) | 2 (1–8) | 2 (1–4) |
| **Hospitalization** | Hospitalized at time of interview | 7 (1.0%) | 6 (75.0%) | <0.001*** | 13 (1.8%) | 27 (90.0%) | 9 (7.3%) |
| | Hospitalized (current phase) | 82 (11.4%) | 6 (75.0%) | <0.001*** | 88 (12.1%) | 27 (90.0%) | 47 (38.2%) |
| | Times hospitalized (current phase) | 1.3 (1.1–1.5) | 1.0 (1.0–1.0) | 0.449 | 1.3 (1.1–1.4) | 1.0 (0.9–1.1) | 1.2 (1.1–1.3) |
| **Mode of TB treatment** | Self administered | 704 (98.2%) | 4 (50.0%) | <0.001*** | 708 (97.7%) | 13 (43.3%) | 107 (87.0%) |
| | Directly observed therapy | 13 (1.8%) | 4 (50.0%) | <0.001*** | 17 (2.3%) | 17 (56.7%) | 15 (12.2%) |
| **Number of health facility visits, mean (95%CI)** | Pre-disease | 1.8 (1.6–2.1) | 2.4 (0.3–4.5) | 0.498 | 1.8 (1.6–2.1) | 1.8 (1.1–2.6) | 1.5 (1.2–1.9) |
| | Directly observed therapy | 2.6 (1.1–4.1) | 136.4 (14.5–258.3) | <0.001*** | 4.1 (2.0–6.2) | 137.1 (85.8–188.4) | 18.6 (8.9–28.3) |
| | Drug pick-up | 16.1 (15.0–17.2) | 15.8 (11.4–20.3) | 0.962 | 15.6 (14.8–16.3) | 18.7 (17.4–20.0) | 12.6 (10.5–14.7) |
| | Follow-up | 2.4 (2.0–2.8) | 0.2 (-0.3–0.6) | 0.274 | 2.4 (2.0–2.8) | 0.8 (-0.4–2.0) | 1.6 (0.8–2.5) |

* Significant difference (0.01 ≤ p < 0.05).

** Significant difference (0.001 ≤ p < 0.01).

*** Significant difference (p < 0.001).

IQR: Interquartile range, 95% CI: 95% confidence interval, DS-TB: drug-susceptible TB, DR-TB: drug-resistant TB.

infected patients and triple for patients with DR-TB (DR-TB: US$ 2,243, TB-HIV: US$ 1,633) compared to DS-TB patients (US$ 748). The high costs among DR-TB and TB-HIV co-infected patients were largely attributed to nutritional supplements outside their normal diet and income loss during TB treatment, while TB-HIV co-infected patients had a relatively large proportion of direct medical costs (DR-TB: nutritional supplement 49.7%, income loss 35.1%) (TB-HIV: nutritional supplement 23.3%, income loss 39.2%, direct medical costs 26.2%) (**Fig 1**).

**Table 3. Time loss for TB care seeking in participants of the national TB patients cost survey by drug resistance and HIV status, Lao PDR, 2018–2019.**

| Time loss due to TB (working hour basis) | | Nationally representative sample | | | | | | | With additional sample | | | |
|---|---|---|---|---|---|---|---|---|---|---|---|---|
| | | DS-TB | | DR-TB | | P-value | | All | | All DR-TB | | All TB-HIV | |
| | | n = 717 | | n = 8 | | | | n = 725 | | n = 30 | | n = 123 | |
| Hours lost by patient, median (IQR) | Overall | 95 | (79–139) | 1,327 | (665–2,517) | 0.006** | 95 | (79–139) | 1,418 | (1,219–2,194) | 122 | (83–220) |
| | Pre-diagnosis | 9 | (3–44) | 40 | (32–270) | 0.050* | 10 | (3–47) | 10 | (4–40) | 24 | (6–53) |
| | Hospitalization | 76 | (37–158) | 755 | (301–1,088) | 0.026* | 78 | (38–165) | 858 | (694–1,088) | 185 | (66–453) |
| | Directly observed therapy | 65 | (30–130) | 39 | (27–45) | 0.306 | 45 | (30–65) | 23 | (23–45) | 15 | (4–30) |
| | Drug pick-up | 9 | (4–17) | 2 | (1–10) | 0.023* | 9 | (4–17) | 2 | (1–3) | 13 | (6–26) |
| | Follow-up | 1 | (0–4) | 0 | (0–4) | 0.291 | 1 | (0–4) | 0 | (0–0) | 0 | (0–0) |
| Hours lost by caregiver, median (IQR) | Overall | 1 | (0–13) | 2 | (0–301) | 0.616 | 1 | (0–13) | 0 | (0–0) | 0 | (0–52) |
| | Hospitalization | 102 | (39–206) | 541 | (61–1,209) | 0.185 | 104 | (42–305) | 964 | (541–1,789) | 337 | (68–765) |
| | Directly observed therapy | 39 | (0–130) | 45 | (45–45) | 0.826 | 42 | (15–97) | 45 | (45–45) | 15 | (7–30) |
| | Drug pick-up | 13 | (5–26) | 21 | (3–39) | 0.935 | 13 | (6–26) | 21 | (3–39) | 13 | (6–26) |
| | Follow-up | 1 | (0–21) | 0 | (0–0) | 0.301 | 1 | (0–21) | 0 | (0–0) | 0 | (0–0) |

* Significant difference ($0.01 \leq p < 0.05$).

** Significant difference ($0.001 \leq p < 0.01$).

IQR: Interquartile range, DS-TB: drug-susceptible TB, DR-TB: drug-resistant TB.

## Reported coping mechanisms and social consequences

In the national sample, half (49.9%) of households relied on savings, loans, and/or selling assets to cope with the financial burden and consequences of TB (Table 5), while this figure was higher for DR-TB patients (56.7%) and TB-HIV co-infected patients (65.0%). One fifth (N = 147, 20.3%) of households experienced food insecurity, 29.5% lost a job, and 10.3% experienced social exclusion due to TB. The proportion of TB-affected households who experienced separation or divorce from their partners or who had interrupted schooling for their children was significantly higher among DR-TB patients (separation/divorce: DS-TB 2.1%, DR-TB 12.5%, p = 0.046) (interrupted schooling: DS-TB 1.4%, DR-TB 12.5%, p = 0.011).

More than half of the national sample (54.5%) reported that TB care had a moderate, severe, or very severe financial impact on their household, and 56.3% perceived that TB treatment impoverished their households, when compared to their baseline financial position (Table 6). These proportions were significantly higher among DR-TB patients in the national sample (financial impact: DS-TB 54.2%, DR-TB 87.5%, p = 0.002. Impoverishment: DS-TB 56.0%, DR-TB 87.5%, p = 0.026).

The proportion of patients who became unemployed more than doubled when comparing the baseline situation to the situation at the time of interview (16.8% to 35.4%), while the proportion of employment in the informal sector decreased from 61.9% to 43.6% (Fig 2). The proportion of formal employment decreased from 10.8% to 8.8% when comparing the same time periods. Only one (0.1%) of the national sample utilized sick leave, and four (0.4%) received social welfare, including an unemployment benefit. Among DR-TB patients, who were eligible to receive support for food and transportation from the NTP, only three (10.0%) reported that they received these TB specific financial supports.

## Proportion of households facing catastrophic costs

In the national sample, the proportion of TB affected households facing catastrophic costs was 62.6% (95% CI 57.6%-67.3%) at the threshold at 20% of annual household income. With

**Table 4. Estimated median total costs incurred by TB-affected households in Lao PDR, 2018–2019, assessed by output approach (in US$).**

| TB patient costs | | | Nationally representative sample | | | | | | With additional sample | | | |
|---|---|---|---|---|---|---|---|---|---|---|---|---|
| | | | DS-TB | | DR-TB | | All | | All DR-TB | | All TB-HIV | |
| | | | median (IQR) | | median (IQR) | | median (IQR) | | median (IQR) | | median (IQR) | |
| | | | n = 717 | | n = 8 | | n = 725 | | n = 30 | | n = 123 | |
| Pre-TB diagnosis | Direct medical | Total | 40 | (6–177) | 71 | (24–73) | 41 | (6–177) | 71 | (24–106) | 138 | (35–353) |
| | | Hospitalization | 4 | (0–31) | 6 | (1–17) | 4 | (0–29) | 2 | (0–11) | 99 | (8–353) |
| | | Outpatient services | 38 | (7–174) | 71 | (13–186) | 38 | (7–174) | 71 | (21–212) | 87 | (12–239) |
| | Direct non-medical | Total | 11 | (4–39) | 40 | (24–52) | 11 | (4–39) | 15 | (9–52) | 17 | (8–79) |
| | | Travel | 4 | (1–12) | 7 | (4–24) | 4 | (1–12) | 9 | (2–13) | 5 | (1–12) |
| | | Food | 1 | (0–8) | 8 | (0–26) | 1 | (0–8) | 1 | (0–5) | 2 | (0–6) |
| | | Accommodation | 0 | (0–0) | 0 | (0–0) | 0 | (0–0) | 0 | (0–0) | 0 | (0–0) |
| Post-TB diagnosis | Direct medical | Total | 27 | (13–30) | 210 | (145–479) | 29 | (13–30) | 0 | (0–83) | 261 | (15–282) |
| | | Hospitalization | 13 | (13–30) | 208 | (111–381) | 13 | (13–30) | 0 | (0–15) | 261 | (15–261) |
| | | Directly observed therapy | 0 | (0–0) | 0 | (0–0) | 0 | (0–0) | 0 | (0–0) | 0 | (0–0) |
| | | Drug pick-up | 0 | (0–31) | 132 | (69–195) | 0 | (0–31) | 69 | (0–195) | 27 | (8–54) |
| | | Follow-up | 0 | (0–3) | 0 | (0–2) | 0 | (0–3) | 0 | (0–0) | 0 | (0–0) |
| | Direct non-medical | Total | 327 | (169–702) | 266 | (209–791) | 327 | (173–703) | 863 | (410–1,276) | 680 | (474–991) |
| | | Travel | 33 | (18–62) | 85 | (78–201) | 33 | (18–64) | 50 | (50–64) | 64 | (33–125) |
| | | Food | 0 | (0–31) | 25 | (18–154) | 1 | (0–31) | 11 | (9–58) | 23 | (12–55) |
| | | Accommodation | 0 | (0–0) | 0 | (0–0) | 0 | (0–0) | 0 | (0–0) | 0 | (0–0) |
| | | Nutritional Supplement | 229 | (104–581) | 161 | (115–210) | 223 | (104–566) | 275 | (183–1,009) | 367 | (229–612) |
| | | Other | 0 | (0–15) | 10 | (0–46) | 0 | (0–15) | 0 | (0–14) | 0 | (0–15) |
| | Indirect costs (output approach) | | 32 | (0–565) | 1,324 | (318–1,881) | 57 | (0–565) | 795 | (0–1,589) | 512 | (0–1,059) |
| **Total direct medical costs** | | | 53 | (53–88) | 280 | (167–526) | 53 | (53–88) | 71 | (21–129) | 261 | (60–440) |
| **Total direct non-medical costs** | | | 338 | (185–735) | 298 | (226–816) | 338 | (186–736) | 868 | (464–1,276) | 696 | (481–1,065) |
| **Total indirect costs** | | | 32 | (0–565) | 1,324 | (318–1,881) | 57 | (0–565) | 795 | (0–1,589) | 512 | (0–1,059) |
| **Total TB patient costs** | | | 748 | (350–1,432) | 2,205 | (1,081–3,273) | 755 | (351–1,454) | 2,243 | (1,231–2,986) | 1,633 | (1,007–2,653) |

IQR: Interquartile range, DS-TB: drug-susceptible TB, DR-TB: drug-resistant TB.

additional sample, the proportion of DR-TB and TB-HIV coinfected patients who experienced catastrophic costs was substantially higher than DS-TB patients; DS-TB: 62.3% (95%CI: 57.4% - 67.0%), DR-TB: 86.9% (73.8% - 99.6%), and TB-HIV: 81.1% (95%CI: 74.1% - 88.2%) (**Fig 3**). The overall incidence of catastrophic costs ranged from 47.3% to 82.2% when changing the threshold from 10% to 30% of annual household income (**Fig 4**).

## Risk factors for households experiencing catastrophic costs

After adjusting for potential confounders and covariates, wealth quintile was the only factor which was associated with the probability of incurring catastrophic costs in multivariate analyses (**Table 7**). Households in lower wealth quintiles had a significantly higher incidence of facing catastrophic costs compared to those in the highest wealth quintile (Lowest wealth quintile: 90.6%, OR = 28.8, p<0.001. 2nd lowest wealth quintile: 73.1%, OR = 6.0, p<0.001. Middle wealth quintile: 56.9%, OR = 3.0, p<0.001. 2nd highest wealth quintile: 53.0%, OR = 2.7, p = 0.001. Highest wealth quintile (Ref): 27.5%).

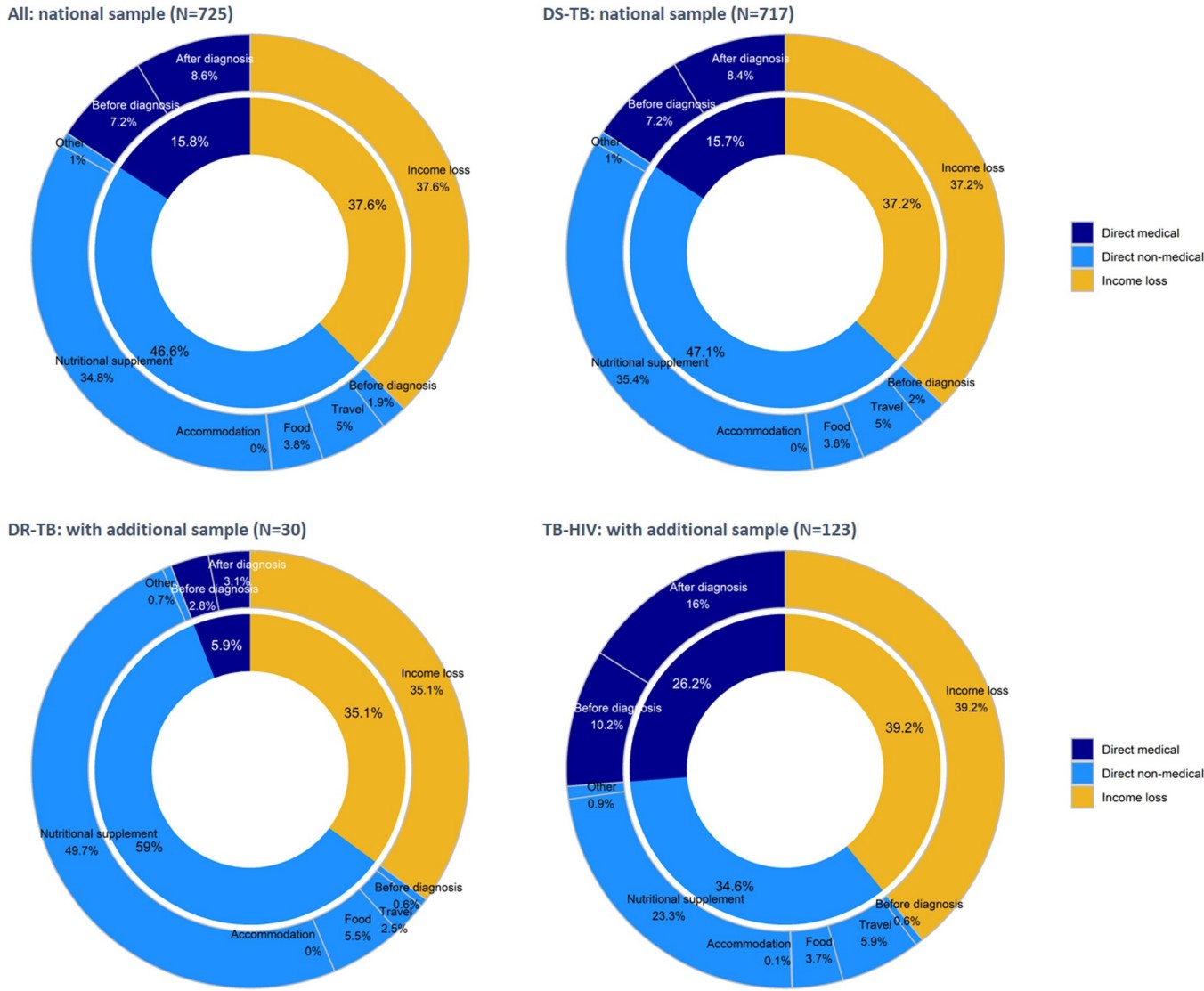

**Fig 1. Composition of TB patient costs in Lao PDR, by drug resistance and HIV status, 2018–2019.**

## Discussion

This survey was the first national TB patient cost survey in Lao PDR. It assessed the magnitude and the main drivers of costs faced by TB patients and their households, designed to aid in the development of policies and interventions to reduce the financial barriers associated with TB care. This survey also establishes a baseline against which to monitor progress towards the elimination of catastrophic costs due to TB in Lao PDR, aligned to the targets in their National TB Strategic Plan and the End TB Strategy [11, 12].

Although free TB services are provided in public health facilities in Lao PDR, 62.2% of DS-TB, 86.7% of DR-TB, and 81.1% of TB-HIV co-infected patients incurred catastrophic costs. A TB diagnosis and care cost households US$ 755 which was equivalent to more than 3 times the average monthly salary of TB patients in the survey. The main cost driver was direct non-medical costs followed by income loss. To cope with the economic burden, half (49.9%) of patients had to rely on savings (21.4%), borrowing money (26.3%) or selling assets (17.7%),

**Table 5. Reported coping mechanisms and social consequences in participants of the national TB patients cost survey by drug resistance and HIV status, Lao PDR, 2018–2019.**

| Coping mechanism and social consequences | | Nationally representative sample | | | | | | With additional sample | | | |
|---|---|---|---|---|---|---|---|---|---|---|---|
| | | DS-TB | | DR-TB | | P-value | All | | All DR-TB | | All TB-HIV | |
| | | n = 717 | | n = 8 | | | n = 725 | | n = 30 | | n = 123 | |
| Coping strategy | Dissaving | 152 | (21.2%) | 3 | (37.5%) | 0.706 | 155 | (21.4%) | 14 | (46.7%) | 59 | (48.0%) |
| | Loan | 189 | (26.4%) | 2 | (25.0%) | 0.610 | 191 | (26.3%) | 7 | (23.3%) | 43 | (35.0%) |
| | Sale of assets | 127 | (17.7%) | 1 | (12.5%) | 1.000 | 128 | (17.7%) | 3 | (10.0%) | 22 | (17.9%) |
| | Any of above | 358 | (49.9%) | 4 | (50.0%) | 0.088 | 362 | (49.9%) | 17 | (56.7%) | 80 | (65.0%) |
| Social effect | Food insecurity | 146 | (20.4%) | 1 | (12.5%) | 0.582 | 147 | (20.3%) | 3 | (10.0%) | 13 | (10.6%) |
| | Divorce/separation | 15 | (2.1%) | 1 | (12.5%) | 0.046* | 16 | (2.2%) | 3 | (10.0%) | 2 | (1.6%) |
| | Job loss | 212 | (29.6%) | 2 | (25.0%) | 0.778 | 214 | (29.5%) | 12 | (40.0%) | 58 | (47.2%) |
| | Interrupted schooling | 10 | (1.4%) | 1 | (12.5%) | 0.011* | 11 | (1.5%) | 1 | (3.3%) | 5 | (4.1%) |
| | Social exclusion | 73 | (10.2%) | 2 | (25.0%) | 0.171 | 75 | (10.3%) | 5 | (16.7%) | 10 | (8.1%) |
| | Any of above | 419 | (58.4%) | 6 | (75.0%) | 0.344 | 425 | (58.6%) | 19 | (63.3%) | 76 | (61.8%) |

* Significant difference (0.01 ≤ p < 0.05).

DS-TB: drug-susceptible TB, DR-TB: drug-resistant TB.

which has the potential to cause prolonged negative impacts on their lives [31]. The key findings of this survey will contribute to open the doors for effective policy dialogues at the national level with multisectoral partners to improve TB service delivery and financing to reduce financial burden due to TB. As of July 2019, 14 countries including Lao PDR had completed a national TB patient cost survey and of those, 8 countries are in Asia (China, Fiji, Lao PDR, Mongolia, Myanmar, Philippines, Timor-Leste, Viet Nam). The proportion of TB-affected households who faced catastrophic costs in this survey was similar to the figures in Mongolia (68%), Myanmar (60%) and Vietnam (63%) [1, 32]. In our survey, a high proportion of catastrophic costs was reported from DR-TB patients and their households, which is consistent with the results from other countries (ranging from 67% to 100%). Similar to the Lao PDR survey, the main contributor to total patient costs was direct non-medical costs in Fiji and Viet

**Table 6. Perceived financial impact and impoverishment in participants of the national TB patients cost survey by drug resistance and HIV status, Lao PDR, 2018–2019.**

| Perceived financial impact and impoverishment | | Nationally representative sample | | | | | | With additional sample | | | |
|---|---|---|---|---|---|---|---|---|---|---|---|
| | | DS-TB | | DR-TB | | P-value | All | | All DR-TB | | All TB-HIV | |
| | | n = 717 | | n = 8 | | | n = 725 | | n = 30 | | n = 123 | |
| Self-reported financial impact | No impact | 79 | (11.0%) | 1 | (12.5%) | 0.002** | 80 | (11.0%) | 4 | (13.3%) | 12 | (9.8%) |
| | Little impact | 250 | (34.9%) | | | | 250 | (34.5%) | 4 | (13.3%) | 30 | (24.4%) |
| | Moderate impact | 222 | (31.0%) | 1 | (12.5%) | | 223 | (30.8%) | 9 | (30.0%) | 30 | (24.4%) |
| | Serious impact | 144 | (20.1%) | 4 | (50.0%) | | 148 | (20.4%) | 10 | (33.3%) | 41 | (33.3%) |
| | Very serious impact | 22 | (3.1%) | 2 | (25.0%) | | 24 | (3.3%) | 3 | (10.0%) | 9 | (7.3%) |
| Self-reported impoverishment | Richer | 0 | (0.0%) | 0 | (0.0%) | 0.026* | 0 | (0.0%) | 0 | (0.0%) | 0 | (0.0%) |
| | Unchanged | 316 | (44.1%) | 1 | (12.5%) | | 317 | (43.7%) | 11 | (36.7%) | 56 | (45.5%) |
| | Poorer | 369 | (51.5%) | 5 | (62.5%) | | 374 | (51.6%) | 13 | (43.3%) | 52 | (42.3%) |
| | Much poorer | 32 | (4.5%) | 2 | (25.0%) | | 34 | (4.7%) | 6 | (20.0%) | 14 | (11.4%) |

* Significant difference (0.01 ≤ p < 0.05).

** Significant difference (0.001 ≤ p < 0.01).

DS-TB: drug-susceptible TB, DR-TB: drug-resistant TB

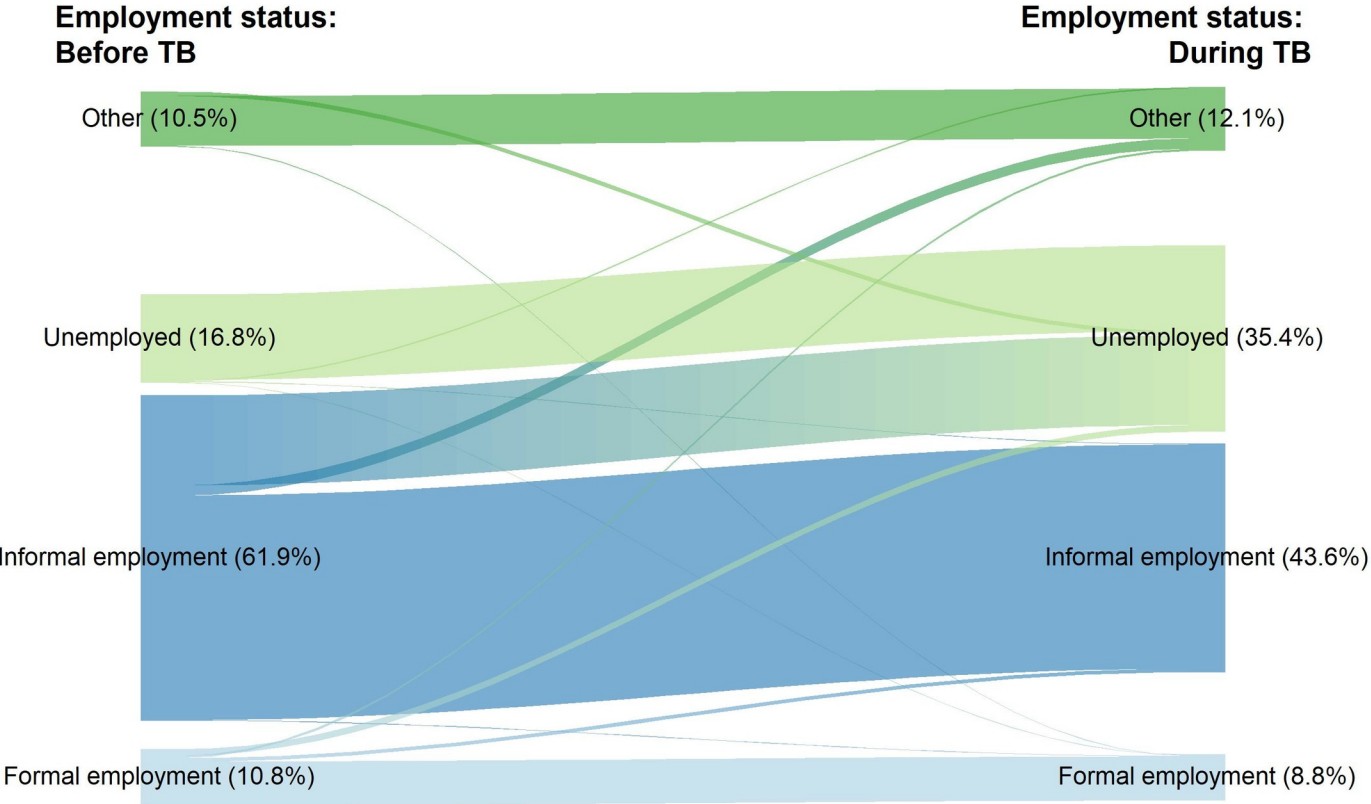

**Fig 2. Changes of employment status: Before having TB and during TB treatment, in national sample (N = 725) of the national TB patients cost survey, Lao PDR, 2018–2019.**

Nam. The survey in Lao PDR was the first national survey to assess costs and catastrophic costs among TB-HIV co-infected patients, and the proportion of TB-affected households who faced catastrophic costs was as high as that of DR-TB. This result has highlighted the necessity to assess patient costs in households with TB-HIV co-infected patients and to facilitate interventions to minimize the financial burden especially in countries with a high burden of TB-HIV coinfection. Integrated services for TB and HIV was provided only at 11 of 25 central and provincial hospitals, and therefore patients with TB-HIV coinfection had to travel to those hospitals that are usually located far from their residences compared to public health centers or district hospitals, or had to have separate facility visits for TB and HIV treatments [26–28]. Enhancing and decentralizing integrated services for TB and HIV would be necessary to mitigate the financial burden in TB-HIV coinfected patients.

Nutritional supplements other than the patient's regular diet comprised (34.8%) of all direct non-medical costs in Lao PDR. Malnutrition is a common clinical finding in TB patients and also a risk factor for developing active TB [33, 34]. Due to a bi-directional association between having TB and malnutrition, TB patients often lose their appetite and body weight when they develop TB, and can then become malnourished due to metabolic changes during TB treatment [35–37]. Patients and their households may believe or be advised that eating protein rich foods enhances the effectiveness of TB treatment leading them to buy additional foods that might lead TB-affected households to spend a relatively large amount on food or other nutritional supplements [38]. In TB patient cost surveys in Kenya and Ghana, body mass index (BMI) of the survey participants was calculated, and more than half of the participants in the

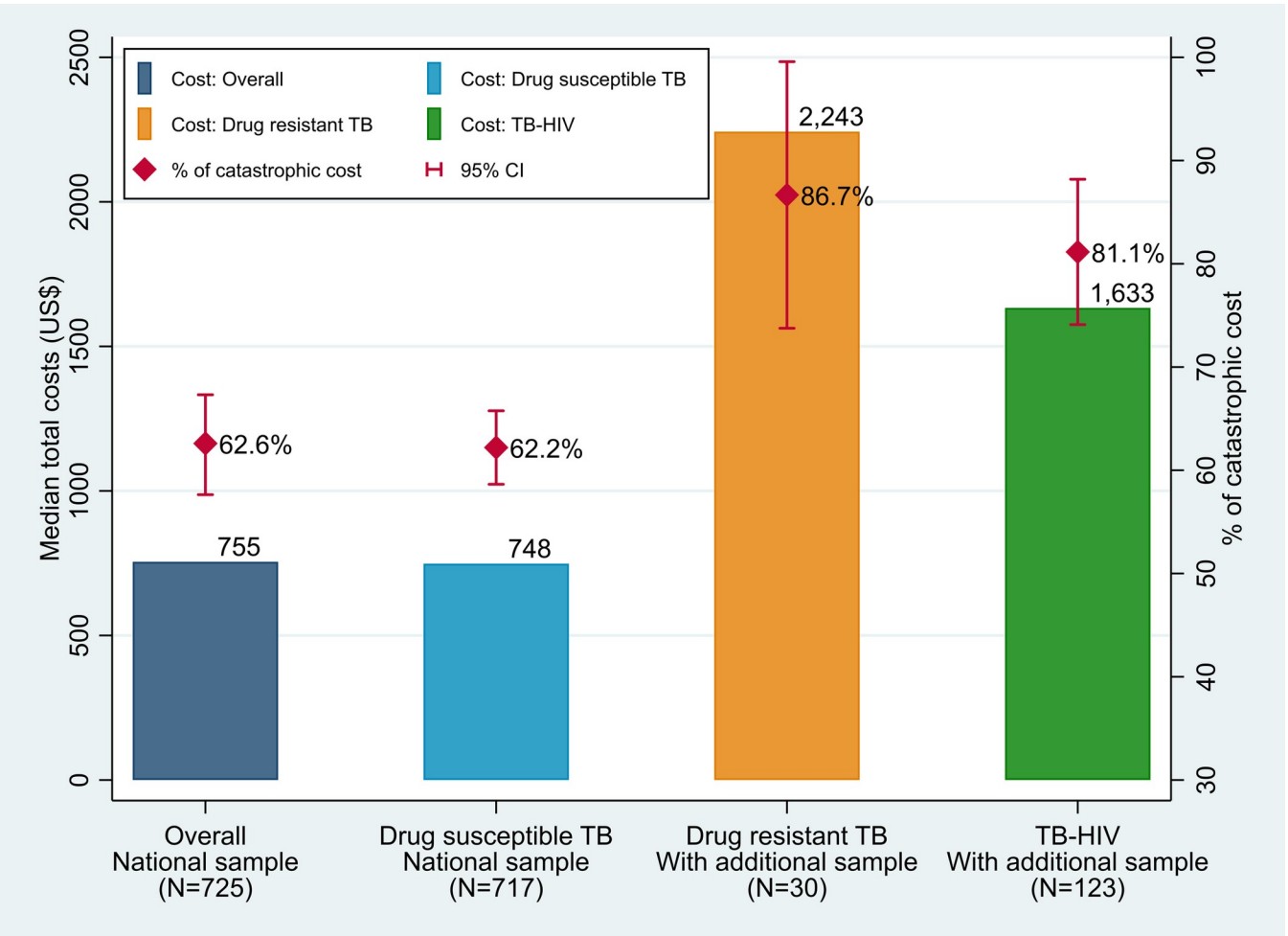

**Fig 3. Median total costs and proportion of households facing catastrophic costs due to TB using 20% threshold of annual household income by drug resistance and HIV status.** * Error bars show 95% confidence interval for proportion of catastrophic costs. ** Overall proportion of catastrophic costs were adjusted for all variables in the final model as well as for clustering effects associated with sampling method.

survey were severely (BMI<16.5) or moderately (BMI<18.5) malnourished, and direct non-medical costs were also a main cost driver as they were in Lao PDR [39]. The results in Kenya and Ghana highlight the necessity to investigate the prevalence of malnutrition and the need to enhance nutritional support for TB patients [39].

In Lao PDR, the NTP implements a cash transfer programme for DR-TB patients that provides US$ 5 per day to support expenses for food and transportation. However, only 10% of the participants with DR-TB in this survey reported that they received such support. Similarly, in Cambodia, an existing government scheme that provides poor people with transportation and food costs associated with their health-seeking to public health facilities was largely under-utilized [40]. Also, it is not known how this cash transfer programme in Lao PDR contributes to TB patients' nutritional recovery. Although WHO recommends systematic nutritional assessment and counselling for TB patients [41], such services are not systematically provided for TB patients in Lao PDR, and nutritional status such as weight, height, and BMI were not investigated in this survey. The National Nutrition Strategy 2016–2020 prepared by the National Nutrition Centre has a strategic objective to prevent TB related malnutrition by managing and controlling acute malnutrition associated with TB [42]. Enhancing collaboration

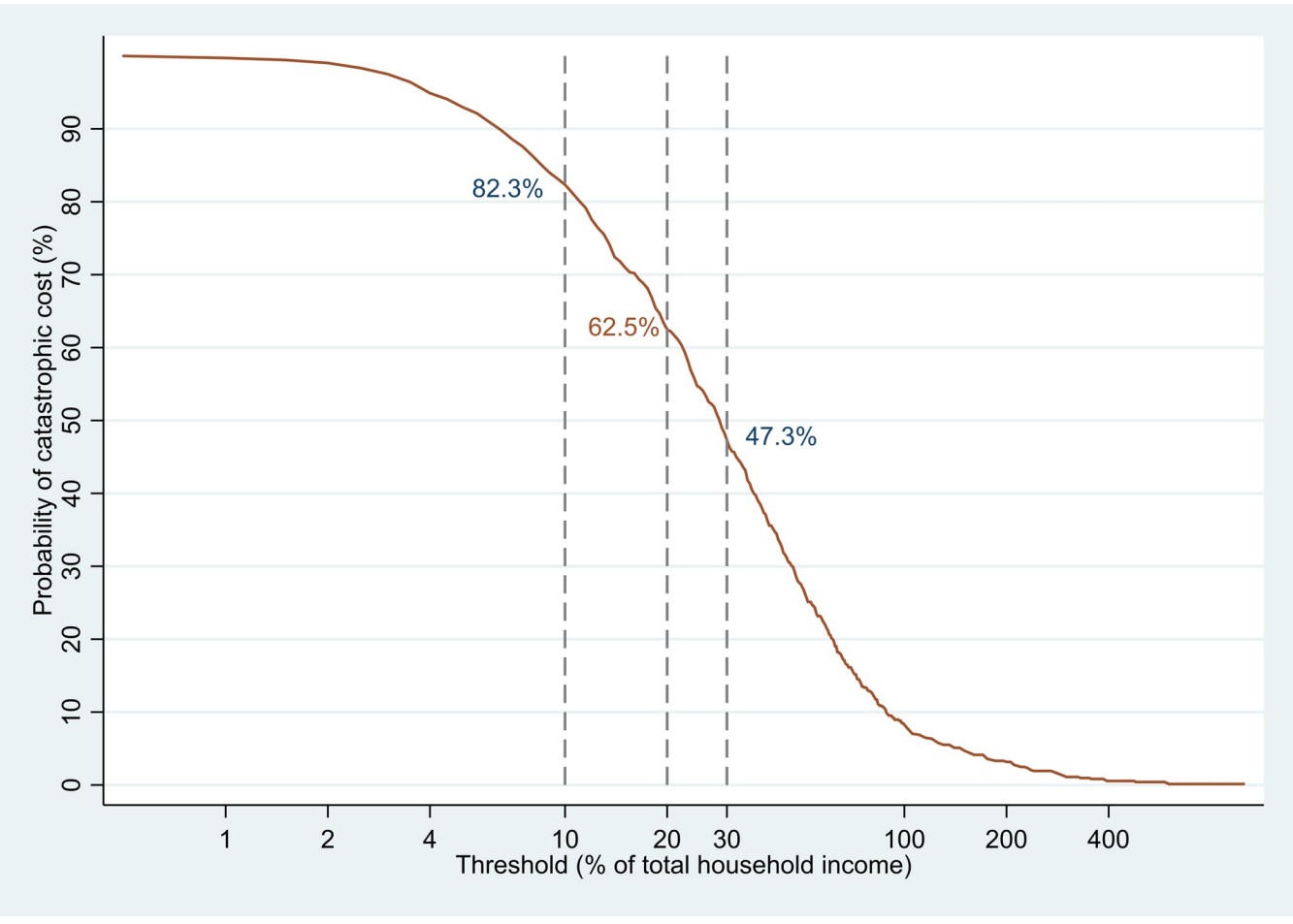

**Fig 4. Changing threshold to define proportion of TB-affected household experiencing catastrophic costs, with national sample (N = 725) of the national TB patients cost survey, Lao PDR, 2018–2019.**

between TB and nutrition programmes is imperative to have a better understanding of TB patient's nutritional status and to improve nutritional support for TB patients.

Although only 9.2% of total costs was incurred before TB diagnosis, this survey revealed that TB patients needed to visit health facilities on average 1.8 times before they were diagnosed, with a median delay of 4 weeks from the onset of TB symptoms to diagnosis. 36% of TB patients sought care at private facilities and half of them visited private facilities multiple times before TB diagnosis. The treatment delay that we observed was in the range of results obtained in other national TB patient cost surveys (0.14 weeks in Timor-Leste, 2.9 weeks in Kenya, 4 weeks in Ghana, 6.2 weeks in Viet Nam) [32, 39, 43, 44] Streamlining the TB patient pathway by enhancing the referral system and improving linkages with the private sector at all facility levels will likely minimize diagnostic delay and patient costs. Implementing active case finding (ACF) may result in a shorter delay in TB diagnosis compared to facility based passive case finding (PCF) as well as reduction in costs incurred by TB patients [45, 46]. A study in Cambodia comparing TB patient costs between ACF and PCF revealed that costs before TB diagnosis was significantly lower among patients detected with ACF compared to those with PCF whereas no difference was found in costs during TB treatment [40]. Furthermore, increasing awareness of the NHI would be also important to have early diagnosis of TB. Our study

**Table 7. Factors associated with catastrophic costs faced by TB-affected households, Lao PDR, 2018–2019.**

| Risk factors | | Total | Facing catastrophic costs | | Univariate | | | Multivariate | | |
|---|---|---|---|---|---|---|---|---|---|---|
| | | | N | % | Crude OR | 95% CI | p-value | Adjusted OR | 95% CI | p-value |
| **Age** | in years | - | - | - | 0.99 | (0.99–1.00) | 0.184 | - | - | - |
| **Sex** | Female | 292 | 187 | (64.0%) | Ref | - | - | - | - | - |
| | Male | 433 | 266 | (61.4%) | 0.91 | (0.64–1.29) | 0.573 | - | - | - |
| **Insurance** | Any insurance | 175 | 105 | (60.0%) | Ref | - | - | - | - | - |
| | No insurance | 550 | 348 | (63.3%) | 1.14 | (0.82–1.59) | 0.415 | - | - | - |
| **Education** | Secondary or higher | 281 | 162 | (57.7%) | Ref | - | - | Ref | - | - |
| | Primary school | 177 | 118 | (66.7%) | 1.42 | (0.87–2.31) | 0.153 | 0.80 | (0.40–1.62) | 0.527 |
| | No education | 267 | 173 | (64.8%) | 1.38 | (0.95–1.99) | 0.088 | 0.93 | (0.55–1.56) | 0.765 |
| **Occupation** | Employed (formal) | 78 | 36 | (46.2%) | Ref | - | - | Ref | - | - |
| | Unemployed | 122 | 72 | (59.0%) | 1.71 | (0.91–3.23) | 0.091 | 0.95 | (0.49–1.85) | 0.870 |
| | Employed (informal) | 449 | 316 | (70.4%) | 2.73 | (1.70–4.39) | <0.001*** | 1.66 | (0.96–2.87) | 0.066 |
| | Other (housework, student etc) | 76 | 29 | (38.2%) | 0.74 | (0.41–1.34) | 0.307 | 0.60 | (0.31–1.16) | 0.122 |
| **Household income quintile** | Highest | 91 | 25 | (27.5%) | Ref | - | - | Ref | - | - |
| | Second highest | 149 | 79 | (53.0%) | 3.06 | (1.78–5.26) | <0.001*** | 2.66 | (1.56–4.52) | 0.001** |
| | Middle | 195 | 111 | (56.9%) | 3.47 | (1.94–6.18) | <0.001*** | 2.98 | (1.72–5.17) | <0.001*** |
| | Second lowest | 141 | 103 | (73.0%) | 7.03 | (3.37–14.65) | <0.001*** | 5.99 | (2.84–12.61) | <0.001*** |
| | Lowest | 149 | 135 | (90.6%) | 29.28 | (13.37–64.13) | <0.001*** | 28.79 | (11.57–71.64) | <0.001*** |
| **Household size** | | - | - | - | 0.95 | (0.89–1.01) | 0.084 | 0.96 | (0.89–1.04) | 0.273 |
| **Treatment phase** | Intensive phase | 270 | 179 | (66.3%) | Ref | - | - | - | - | - |
| | Continuation phase | 455 | 274 | (60.2%) | 0.76 | (0.51–1.14) | 0.175 | - | - | - |
| **Drug susceptibility** | Drug susceptible TB | 717 | 446 | (62.2%) | Ref | - | - | Ref | - | - |
| | Drug resistant TB | 8 | 7 | (87.5%) | 4.02 | (0.47–34.56) | 0.194 | 3.89 | (0.57–26.29) | 0.156 |
| **Treatment category** | New | 710 | 444 | (62.5%) | Ref | - | - | - | - | - |
| | Relapse | 12 | 8 | (66.7%) | 1.34 | (0.32–5.59) | 0.680 | - | - | - |
| | Loss to follow-up or Treatment after failure | 2 | 1 | (50.0%) | 0.53 | (0.03–10.28) | 0.665 | - | - | - |
| **Delay in TB diagnosis** | 1 month or less | 627 | 391 | (62.4%) | Ref | - | - | - | - | - |
| | More than 1 month | 98 | 62 | (63.3%) | 1.05 | (0.62–1.77) | 0.859 | - | - | - |
| **HIV status** | Negative/unknown | 706 | 440 | (62.3%) | Ref | - | - | - | - | - |
| | Positive | 19 | 13 | (68.4%) | 1.38 | (0.47–4.07) | 0.546 | - | - | - |
| **Self reported financial impact** | No or little impact | 330 | 163 | (49.4%) | Ref | - | - | - | - | - |
| | Moderate, serious, very serious impact | 395 | 290 | (73.4%) | 2.89 | (1.96–4.28) | <0.001*** | - | - | - |
| **Self reported impoverishment** | Richer/unchanged | 317 | 145 | (45.7%) | Ref | - | - | - | - | - |
| | Poorer/much poorer | 408 | 308 | (75.5%) | 3.66 | (2.48–5.41) | <0.001*** | - | - | - |

* Significant difference (0.01 ≤ p < 0.05)

** Significant difference (0.001 ≤ p < 0.01)

*** Significant difference (p < 0.001)

95% CI: 95% confidence interval, OR: Odds Ratio

Crude ORs are adjusted for clustering effects associated with sampling method

Adjusted ORs are adjusted for all variables in the final model as well as for clustering effects associated with sampling method

showed a considerably low recognition of NHI coverage among TB patients (75.9% reported no insurance) even though the NHI was implemented in 2016 (2 years back from the time of this study). This low recognition of NHI could be a barrier to use healthcare services in public facilities after having TB symptoms.

Currently, free TB services are provided in public health facilities in Lao PDR with financial support from the Global Fund in addition to government funding, and therefore the direct medical costs had a relatively smaller impact on total TB patient costs in this survey (15.8%). Ensuring the future sustainability of free and high-quality TB services in this country is key to minimize the delay in TB diagnosis and initiating TB treatment and to reduce the out-of-pocket expenditure for the direct medical costs for TB.

The proportion of patients who were unemployed increased when comparing patient's employment status pre and post TB diagnosis. Patients who were working in the informal sector, which consisted of 61.9% of the survey population, were more likely to have lost their jobs, and currently there is no unemployment protection scheme for those working in the informal sector in Lao PDR. Implementing an additional cash transfer programme would be one option to address this issue. In Vietnam, after conducting TB patient cost survey, the NTP implemented an innovative financial support scheme, so called "Patients Support Foundation to End Tuberculosis (PASTB)" for TB patients in poverty using a short message service (SMS) [47]. In this campaign, every SMS will transfer US$0.8 to the foundation to provide financial support for TB patients [47]. In addition to the financial supports during TB treatment, the post-treatment socio-economic recovery (e.g. re-employment after TB treatment) is also important to minimize long-term or permanent financial impacts due to TB. However, no studies assessing long-term economic shocks in TB-affected households after completion of TB treatment (e.g. permanent job loss, continuous social exclusion) are published yet except for an on-going study in African countries [48]. Therefore, more evidences around post-treatment economic impacts due to TB are required especially in Asian contexts. On the other hand, TB patients who are employed in the formal sector are eligible for unemployment and sickness benefits that are provided by NSSF. The sickness benefit covers 70% of the employee's salary in the first 6 months of sick leave, and the rate decreases to 60% after the first 6 months [22]. However, only one TB patient in this survey received this sickness benefit. The current claim mechanism requires TB patients or their household members to visit district or provincial offices that are usually located far from the patient's community. Re-designing the claim mechanism in collaboration with NSSF may facilitate wider access to the sickness benefit, thus minimizing TB patient costs.

In addition, the national HIV/AIDS Control and Prevention law implemented since 2010, prohibits discrimination or stigmatization towards people living with HIV/AIDS, including firing healthy HIV positive persons from their jobs [49]. There may be a need to explore collaboration with the labour and corporate sectors, as well as civil society, to investigate workplace policies and to advocate for stronger legal frameworks such as enactment of a TB law, with provisions to protect people with TB from being fired, such as the case with the HIV law.

This survey has several limitations. First, this survey was a cross-sectional study and the estimated total costs were based on an extrapolation method [12]. Longitudinal methods which incorporate information from multiple interviews with each patient over time may reflect the true costs of TB illness, although this method would be more complex and lengthy. Estimated costs incurred by TB patients and their households might be affected by recall bias if only recalled once and when recalling costs that were incurred in the past. Also, some of the total patient costs were extrapolated from the costs incurred by patients in the intensive phase or those in continuation phase. This crude extrapolation method may result in over or under estimation of costs and the overall incidence of catastrophic costs. Second, 123 additional

patients (22 DR-TB patients and 101 TB-HIV coinfected patients) were enrolled with purposive sampling at different time (in May-June 2019) separately from the 725 nationally representative sample (in December 2018-January 2019). Thus, statistical comparison between two samples were not carried out for this reason. Furthermore, although we conducted statistical tests comparing various factors between DS-TB and DR-TB patients in the national sample, the number of DR-TB patients in the national sample was very few (N = 8) and not enough especially for risk factor analysis for facing catastrophic costs. Third, study participants were enrolled only from NTP engaged facilities. Therefore, costs incurred by patients receiving TB treatment in private facilities unlinked to the NTP are not captured in this survey, and those patients may have different socio-demographic characteristics and may face different costs. In Lao PDR, however, there was only a limited number of private health facilities, and all the individuals with suspected TB who were identified in private facilities should be referred to public health facilities and initiate TB treatment in public facilities. Fourth, the sampled facilities were randomly selected based on the number of TB case notifications in 2017. This sampling method tended to not select districts with a small number of TB case notifications where the accessibility to healthcare services might be limited. The findings of this survey may underestimate costs due to limited sampling of the population who have issues in accessing health care. Fifth, this survey assessed costs incurred from the time of symptom onset to the end of TB treatment. Therefore, costs after TB treatment has finished were not included.

## Conclusions

The results of the survey showed that although TB diagnosis and treatment are provided free of charge in Lao PDR, TB patients and their households incur substantial costs when they are diagnosed with TB and they also lack financial protection. As non-medical and indirect costs accounted for more than 80% of the total costs, providing free TB services is not enough, and expansion of existing social protection mechanisms and/or implementation of new interventions for TB patients are necessary to mitigate this financial burden and reduce the proportion of households who experience catastrophic costs associated with TB.

## Supporting information

**S1 Questionnaire. Survey instrument for Lao PDR National TB patient cost survey.**
(PDF)

**S1 Table. List of selected provinces and number of clusters for a tuberculosis patient cost survey in Lao PDR.**
(DOCX)

**S2 Table. Post-hoc analysis of the incurrence of catastrophic costs by treatment facility.**
(DOCX)

**S1 Text. Types of household assets used for imputing household income and the proportion of participants for whom imputed income had to be employed.**
(DOCX)

## Acknowledgments

First, we would like to thank the TB patients who consented to participate in this first national TB patient cost survey in Lao PDR. Also, we are grateful for the support from the Ministry of Health and staff at the Central, Provincial or District levels, particularly those from the

Department of Communicable Diseases Control, the Provincial TB Coordinators, the District TB Managers, the interviewers and the health care workers in Lao PDR.

## Author Contributions

**Conceptualization:** Phonenaly Chittamany, Takuya Yamanaka, Sakhone Suthepmany, Thipphasone Vixaysouk, Fukushi Morishita, Nobuyuki Nishikiori.

**Data curation:** Takuya Yamanaka, Sakhone Suthepmany, Thepphouthone Sorsavanh, Phitsada Siphanthong, Vilath Seevisay.

**Formal analysis:** Takuya Yamanaka, Fukushi Morishita.

**Funding acquisition:** Thipphasone Vixaysouk, Fukushi Morishita, Nobuyuki Nishikiori.

**Investigation:** Sakhone Suthepmany, Thepphouthone Sorsavanh, Phitsada Siphanthong, Vilath Seevisay.

**Methodology:** Takuya Yamanaka, Kerri Viney, Thipphasone Vixaysouk.

**Project administration:** Thepphouthone Sorsavanh, Phitsada Siphanthong, Thipphasone Vixaysouk, Moeko Nagai, Vilath Seevisay, Kiyohiko Izumi.

**Resources:** Takuya Yamanaka, Kerri Viney, Thipphasone Vixaysouk, Moeko Nagai, Kiyohiko Izumi.

**Software:** Takuya Yamanaka.

**Supervision:** Phonenaly Chittamany, Sakhone Suthepmany, Jacques Sebert, Thipphasone Vixaysouk, Moeko Nagai, Kiyohiko Izumi, Fukushi Morishita, Nobuyuki Nishikiori.

**Validation:** Takuya Yamanaka, Vilath Seevisay, Kiyohiko Izumi.

**Visualization:** Takuya Yamanaka.

**Writing – original draft:** Takuya Yamanaka.

**Writing – review & editing:** Phonenaly Chittamany, Sakhone Suthepmany, Thepphouthone Sorsavanh, Phitsada Siphanthong, Jacques Sebert, Kerri Viney, Thipphasone Vixaysouk, Moeko Nagai, Kiyohiko Izumi, Fukushi Morishita, Nobuyuki Nishikiori.

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
