## [Decision Letter · Decision Letter 0]

30 Sep 2020

PONE-D-20-27870

First national tuberculosis patient cost survey in Lao People’s Democratic Republic: Assessment of the financial burden faced by TB-affected households and the comparisons by drug-resistance and HIV status

PLOS ONE

Dear Dr. Yamanaka,

Thank you for submitting your manuscript to PLOS ONE. After careful consideration, we feel that it has merit but does not fully meet PLOS ONE’s publication criteria as it currently stands. Therefore, we invite you to submit a revised version of the manuscript that addresses the points raised during the review process.

This is an important piece of operational research, which adds to the growing body of evidence on the huge socioeconomic impact of TB, especially in low-income countries. However, there are some limitations that require addressing prior to re-submission.

Please address Reviewer 1 and 2's comments with special focus on:

1) The criteria for selection and the statistical robustness/limitations of the chosen analyses of the population affected by DR-TB and HIV-TB (please also indicate/consider involvement of a trained statistician)

2) Clarification on data collected including hospitalisation costs, lost income, and carer/time costs, and a description of how missing data was handled

3) Enhancing your description of regression analysis including whether there was adjustment for clustering / intra-cluster coefficients

We look forward to receiving a revised version of the manuscript for further review.

We look forward to receiving your revised manuscript.

Kind regards,

Tom E. Wingfield

Academic Editor

PLOS ONE

Journal Requirements:

3. Please describe in further detail, how you accounted for clustering in your statistical analysis and aslso discuss whether you applied survey weights to ensure that estimates were nationally representative.

4.We note that you have indicated that data from this study are available upon request. PLOS only allows data to be available upon request if there are legal or ethical restrictions on sharing data publicly. For information on unacceptable data access restrictions, please see http://journals.plos.org/plosone/s/data-availability#loc-unacceptable-data-access-restrictions.

5. We note you have included a table to which you do not refer in the text of your manuscript. Please ensure that you refer to Table 3 in your text; if accepted, production will need this reference to link the reader to the Table.

Reviewers' comments:

Reviewer's Responses to Questions

**Comments to the Author**

1. Is the manuscript technically sound, and do the data support the conclusions?

Reviewer #1: Yes

Reviewer #2: Yes

2. Has the statistical analysis been performed appropriately and rigorously? 

Reviewer #1: Yes

Reviewer #2: No

3. Have the authors made all data underlying the findings in their manuscript fully available?

Reviewer #1: Yes

Reviewer #2: No

4. Is the manuscript presented in an intelligible fashion and written in standard English?

Reviewer #1: Yes

Reviewer #2: Yes

5. Review Comments to the Author

Reviewer #1: Review: PONE-D-20-27870

Introduction.

1. The introduction section is still too long. I suggest authors to shorten this part and directly introduce the Lao, a country that is the focus area of this paper instead of bringing up the Lao part at the end of the intro section.

2. Paragraph 5 (Lines 70-78). This para is more likely the basis for discussion, and is redundant later in discussion part. I suggest to remove this para from intro section and elaborate the data in discussion section.

Methods

3. Study setting. Authors may shorten this part, but explain more on how TB services are delivered in the Lao. What are facilities that deliver the services (public only, or also private; primary care level or only secondary/hospital)? What is the link between facilities and the NTP or NHI? What are cost items covered by the NTP and the NHI; any differences? Self-administered/Directly observed therapy?

4. Sampling (lines 134-135). “..,we enrolled additional and operationally feasible quotas of 120 TB-HIV co-infected patients and 30 DR-TB (i.e. MDR-TB or RR-TB).” What is the basis of this sampling size and quota?

5. Is it not clear if the 25-cluster was randomized from a number of clusters, or it is a total number of clusters in the Lao? How were subjects enrolled (randomly, consecutively, etc)?

6. Inclusion (lines 139-142). Why did authors include children in this study? The TB diagnostic tests for children are different with those for adults and unspecific symptoms and signs occur in many cases. These may lead to a higher cost for diagnosis. They are also more likely to having no job, then we can’t see patients’ financial burden caused in this group. The proportion is very few (0.8%). Is it possible to rule out this group? If it is not possible, authors need to provide a reasonable basis in the method section, analyze the differences between children and adults, and discuss it later in discussion section.

Results

7. The number of DR-TB patients in national representative sample is very few (n=8). We then see some cells are empty (e.g., occupation, insurance type, facility type). In Table 3, only one DR-TB was included in analysis. Are they sufficient number for authors to do inference analysis to see the difference between DS-TB and DR-TB? Please consult to statistician regarding this issue: what is the best way to display it fairly.

8. Authors used clustered sampling method. Do authors need to adjust with the sampling method by using random effects and, for example, generalized linear mixed model for analyses? Please consult to statistician regarding this issue.

9. It seems that nutritional supplement played a significant role in costs incurred by patients. Authors need to define ‘nutritional support’ in methods section or below the table so that readers can easily understand what this term means.

10. Table 4. Why did not authors display costs incurred by guardian(s)? Also, indirect costs in pre-diagnosis phase: are they missed?

11. Table 7. Put ‘Ref’ or ‘1’ in OR column for reference variables.

12. Table 7. Authors analyzed determinants of catastrophic costs for mixed group (DS and DR-TB). The costs in DR-TB group were much higher than the costs incurred by DS-TB group. Why did authors mix the groups in the analyses instead of separate the groups? Since the number of DR-TB subjects are very few, it may be enough to analyze it in descriptive way rather than using regression? Please consult statistician regarding this issue.

Discussion

13. Lines 368: “In Lao PDR, the NTP implements a cash transfer programme for DR-TB patients that provides US$ 5 per day to support expenses for food and transportation.” Was the cash transfer included in costs calculation? Please also confirm it in method section including how the transfer is delivered (condition, unconditional, etc.).

Additional

14. Please avoid using number at the first word in sentence.

15. The paper readability is acceptable, but still need further English language editing.

Reviewer #2: Summary of the study

This study reported the results of the first national patient cost survey conducted in Lao PDR. The survey was also the first survey to assess the rate of catastrophic cost incurrence among TB-HIV patients. The study found that at least two-thirds of surveyed TB-affected families suffered catastrophic costs (higher in MDR-TB and TB-HIV patients), which was in line with comparator countries in the region. It concludes with the need for implementation of interventions, social protection mechanisms and policy development to achieve the WHO End TB Strategy of zero TB-affected families suffering from catastrophic costs.

Overall assessment

This manuscript presented highly relevant results that should be published and will contribute to the evidence base on incurred cost per episode of TB in Lao. It was very well written with only minor spelling and grammatical issues as well as the occasional extraneous space. There are a number of concerns and issues that will require explanation and revision of the manuscript as outlined below. Once these issues are satisfactorily addressed, the manuscript can be approved for publication.

Detailed comments

Introduction

- Line 80: Please describe any prior costing work done in Laos regarding healthcare in general and for TB in particular, if any, besides using the WHO patient cost survey tool.

Methods

- Line 132: Please add text to justify the selection of 50% catastrophic cost incurrence in your sample size estimate.

- Line 133: Please describe the reason for separating the initial nationally representative sample and the additional sample, and particularly please describe whether the 30 MDR-TB patients from the additional sampling were included in the DS-TB vs MDR-TB comparisons as well as the regression analysis. If they were excluded, please explain why.

- Lines 139-143: Please describe the standard treatment length of MDR-TB in Laos and among the MDR-TB cases included on the survey. Especially note if any patients were on shortened MDR-TB regimen. Please do the same for extra-pulmonary TB and provide the average/median treatment duration for the 52 EP-TB patients in the study.

- Line 158: It is not clear how conducting each survey only once minimizes recall bias. Instead, a longitudinal survey may have been the more appropriate way to minimize recall bias. As such, perhaps remove the phrase “To minimize recall bias.”

- Lines 159-160: Presumably you asked about income prior to the current episode of TB (as per WHO patient cost tool) along with current income, so it is probably best to list that here as well.

- Lines 161-162: Please describe how you treated caregiver time loss and costs, i.e., were these included in the cost calculation.

- Lines 161-162: Please mention and explain further how time loss was translated into monetary value (i.e., human capital approach, minimum hourly wage...).

- Lines 161-162: Please describe how you treated caregiver time loss and costs, i.e., were these included in the cost calculation.

- Lines 162-164: Since pre-treatment costs were not asked of participants in the CP (62.8% of participants), clearly describe the assumptions made in the extrapolation of pre-treatment costs for the whole sample.

- Lines 187-188: Please describe how you treated missing values in the responses (besides the imputed household income).

- Lines 187-188: Please describe how you treated hospitalization costs and whether you were able to track any hospitalization costs after the survey was performed (but before the end of treatment).

- Lines 187-188: If household assets were used for imputation of HH income, please furnish supplemental information on the types, frequencies and regression results for the imputed HH income as well as the proportion of participants for whom imputation had to be employed

- Lines 187-188 & 194-198: Did your regression analysis for imputation of HH income and for evaluation of association of patient covariates with catastrophic cost incurrence account for clustering effects and intra cluster correlation from your sampling strategy? If it did, please describe in further detail in the paper the regression methods used. If not, please re-fit your model using appropriate specifications including a description of how you arrived at the final model specifications or explain why you chose not to account for ICC

- Line 195-196: Please describe and provide examples how including the additional sample of MDR-TB and TB-HIV participants would have affected the regression analysis.

- Line 317: Please describe wealth quintile in this section rather than the results and elaborate on what "based on household income" means. If it is simply quintiles of Household income then please explain and label it so, since wealth can include other tangible and intangible factors.

Results

- Lines 216-217: It is very surprising to see the proportion of participants without health insurance reported as 75.9% when the national statistics claim that insurance coverage is 60%, 70% or even 79.3% in 2019. Please furnish the Insurance coverage rate among TB patients notified by the Lao NTP to understand if there is any bias in the study sample.

- Lines ibid: If the average national insurance coverage among TB patients is really as low as found in the patient first survey, please include this fact in the study setting section.

- Line 228: The proportion of TB patients in the results treated at public health and district facilities states 61% for the national sample, while the table shows 71%. Please review all of your numbers again and ensure there is internal consistency.

- Lines 228-229: Please provide a post-hoc analysis (as supplemental information) of the CC incurrence in the subpopulation of the 29.9% of participants treated at the provincial/national facilities. For this post-hoc analysis, test whether the subpopulations are significantly different from each other.

- Lines 239-240: Given that 97.7% of the national sample was treated under SAT, please provide information on treatment monitoring standards under national TB treatment guidelines in the methods. Please discuss and clarify the root cause behind the high rate of income loss under SAT.

- Line 580: Please describe the discrepancies in sample sizes between the total participant figure (n=725) in the national sample and respondents for time loss due to TB (n=292).

Discussion

- Line 337: please reference any sources that have formally assessed and linked dissavings and asset sale to prolonged negative impact on their lives.

- Line 391: Please rephrase the word minimal perhaps as 16% of one's income is not necessarily minimal, especially for low income households.

- Line 392: "Ensuring the future sustainability of free and high-quality TB services..." is perhaps more of a basic essential for a TB program rather than a strong discussion point. Sustained free and high-quality TB services is the bare minimum expectation and without it, there should be no End TB Strategy. As this paragraph talks about diagnostic delay and direct medical costs, perhaps identify interventions from the literature that have shown to reduce these barriers to care. One such place could be line 389, which should include an example or at minimum a few references.

- Line 398: The discussion of informally employed participants seems insufficient and a disproportional amount of text is dedicated to formally employed persons who comprise less than 10% of the sample. Hence, please expand the discussion on informally employed participants. Specifically, in the absence of any unemployment protection schemes, which in LMIC often only apply after having paid into the scheme through formal employment, the question for informally employed persons is how fast they were able to recover and pick up their employment again. Informal employment usually has lower barriers to entry into the job market (e.g., rag picking, selling lottery tickets, street-side parking attendants, etc), so perhaps the discussion can allude to the current lack of evidence on post-treatment socioeconomic recovery.

- Lines 175-177 & Lines 425-428: The reasoning for not conducting statistical analysis between DS-TB patients (n=717) and TB/HIV patients (n=123) or even MDR-TB patients (n=30) when you performed statistical comparisons between DS-TB and MDR-TB (n=8) within the national sample is not clear. The reasoning for not performing statistical comparisons between the methods and discussion also seem discordant. As such, please rephrase to provide a more plausible reason for the lack of statistical comparison with the additional sample.

- Line 427: Please remove the word "due".

- Line 432: Please change "suspected TB cases" to persons with presumptive/suspected TB to avoid the use of stigmatizing language.

- Line 434: "Therefore, we assumed that the impact of this exclusion criteria would be minimal." - to be able to make this assumption, it would be necessary to have information on the proportion of missed cases and proportion treated in the private sector (onion model). Please furnish this information or consider rephrasing to not state a potentially false assumption.

- Lines 435-436: this is a very good and critical point. Please provide a map/table in the supplemental information of sample sizes by geography to identify any potential source for bias.

6. PLOS authors have the option to publish the peer review history of their article (what does this mean?). If published, this will include your full peer review and any attached files.

Reviewer #1: No

Reviewer #2: No

---

## [Author Response · Author response to Decision Letter 0]

13 Oct 2020

Response to Reviewers’ Comments:

Thank you for your positive response to our journal submission. We appreciate the constructive feedback and have strove to incorporate your feedback into our journal article. The itemized responses are as follows:

Reviewer 1

1 The introduction section is still too long. I suggest authors to shorten this part and directly introduce the Lao, a country that is the focus area of this paper instead of bringing up the Lao part at the end of the intro section.

 Thank you for your suggestion. We deleted para 2 and para 5 

2 Paragraph 5 (Lines 70-78). This para is more likely the basis for discussion, and is redundant later in discussion part. I suggest to remove this para from intro section and elaborate the data in discussion section.

 Since the discussion already has similar statements, we simply deleted the para 

3 Authors may shorten this part, but explain more on how TB services are delivered in the Lao. What are facilities that deliver the services (public only, or also private; primary care level or only secondary/hospital)? What is the link between facilities and the NTP or NHI? What are cost items covered by the NTP and the NHI; any differences? Self-administered/Directly observed therapy?

 Thank you for the comment. In Lao PDR, TB treatment is predominantly provided at public health facilities. For DS-TB patients, the treatment is provided in 164 TB units including 5 central hospitals in Vientiane Capital, 17 provincial hospitals and 142 district hospitals with self-administered therapy while all the DR-TB patients are hospitalized at 3 provincial hospitals with PMDT throughout the treatment period. 

TB diagnosis and treatment in health facilities is supported by NTP free of charge to patients including diagnostic tests (direct microscopy and molecular test) and first and second-line anti-TB medications. If available, health insurance schemes (NHI) may cover some other services (e.g. hospitalization fees for DS-TB patients). 

Besides, NTP supports DR-TB patients for transportation to health facility, hospitalization for the full treatment duration, laboratory examinations (e.g. LPA second line, initial culture, EKG, biological tests) before treatment start and for treatment response monitoring, adverse drug event (ADE) prevention and monitoring, ancillary medicines and daily food allowance.

4 Sampling (lines 134-135). “..,we enrolled additional and operationally feasible quotas of 120 TB-HIV co-infected patients and 30 DR-TB (i.e. MDR-TB or RR-TB).” What is the basis of this sampling size and quota?

 As set by the WHO End TB Strategy, countries have to report the proportion of TB patients who faced catastrophic costs due to TB through a survey enrolling a nationally representative sample of TB patients. Our main survey sample was designed to produce this indicator for the global monitoring and the data collection was conducted from December 2018 to January 2019. 

Apart from this nationally representative sample, the National TB Progamme of Lao PDR had a keen interest in having additional samples to understand the economic burden incurred by TB patients who received treatment for drug-resistant TB, as well as those having TB-HIV coinfection. For DR-TB patients, we enrolled all the patients on DR-TB treatment at the time of interview in the country (total sampling). For TB-HIV patients, assuming estimated proportion of catastrophic costs: 80% with 10% precision and design effect of 2, the sample size was estimated at 122. We added this sentences in the method section. 

5 Is it not clear if the 25-cluster was randomized from a number of clusters, or it is a total number of clusters in the Lao? How were subjects enrolled (randomly, consecutively, etc)?

 Out of 165 BMUs in Lao PDR, a total of 25 BMUs (clusters) was randomly selected by a probability proportional to size (PPS) method applying the TB case notification in 2017 for each BMU. The participants were enrolled also randomly from TB log-book at each facility.

6 Inclusion (lines 139-142). Why did authors include children in this study? The TB diagnostic tests for children are different with those for adults and unspecific symptoms and signs occur in many cases. These may lead to a higher cost for diagnosis. They are also more likely to having no job, then we can’t see patients’ financial burden caused in this group. The proportion is very few (0.8%). Is it possible to rule out this group? If it is not possible, authors need to provide a reasonable basis in the method section, analyze the differences between children and adults, and discuss it later in discussion section.

 This study followed the WHO methodology to measure the catastrophic costs due to TB defined by the WHO End TB Strategy. The WHO methodology recommends including patients in all age groups in the sampling frame including children. 

This is because the survey is intended to measure economic impact at the household level regardless of whether a patient is adult or child. By using the same instrument, interviews are conducted with guardians for child patients. All income and cost information are relevant when measured at the household level (e.g. costs incurred for the care for their children, lost incomes/jobs, and coping mechanisms in their households) and consequences can be serious because the guardians have to put time and resources for the care of the children. As rightly pointed out, the proportion of childhood patients was small (0.8%) but we believe it is appropriate to keep these observations as per the WHO methodology. 

7 The number of DR-TB patients in national representative sample is very few (n=8). We then see some cells are empty (e.g., occupation, insurance type, facility type). In Table 3, only one DR-TB was included in analysis. Are they sufficient number for authors to do inference analysis to see the difference between DS-TB and DR-TB? Please consult to statistician regarding this issue: what is the best way to display it fairly.

 Thank you for your suggestion.

First, we apologize there were errors in table 3. The number of patients for the analysis was 717 DS-TB and 8 DR-TB same as in other tables. We corrected the error. 

Regarding the statistical analysis, considering the small sample size, we used Fisher’s exact test for comparing categorial variables. Even with a limited sample size for DR-TB patients, some categories in table 2,3,5,6 showed significant differences between DS-TB and DR-TB. Therefore, we would like to keep the analytical method as it is currently shown. 

As rightly pointed out, the DR-TB group (N=8) of the main sample may not be appropriately powered to make statistical inference. As mentioned above (No.4), the purpose of this sample is to serve as a nationally representative sample, rather than (not primarily for) disaggregated analysis. Therefore, we agree that there is a limitation on to what extent we could look into details of the result especially for DR-TB group (in a sense, the disaggregation of national sample is more for looking into DS-TB group). 

To compliment this limitation, the study included the additional samples of DR-TB (N=30) and TB-HIV coinfected patients (N=123) and presented in the same table where relevant. 

We will revise relevant parts of text to incorporate above points. 

8 Authors used clustered sampling method. Do authors need to adjust with the sampling method by using random effects and, for example, generalized linear mixed model for analyses? Please consult to statistician regarding this issue.

 Thank you for your suggestion. We took the adjustment for clustering into account in the analysis. Considering changes in ORs with the adjustment, we changed the table 7 with results with the adjustment for sampling method. Also, after adjusting sampling method, the overall % of catastrophic costs was changed to 62.6% (0.1% increase from the previous result). We also changed figure 2 accordingly.

9 It seems that nutritional supplement played a significant role in costs incurred by patients. Authors need to define ‘nutritional support’ in methods section or below the table so that readers can easily understand what this term means.

 Thank you for your suggestion. We added a definition of nutritional supplements in the method section as “nutritional supplements such as vitamin supplements and/or additional foods other than regular diets”.

10 Table 4. Why did not authors display costs incurred by guardian(s)? Also, indirect costs in pre-diagnosis phase: are they missed?

 In the data collection, we asked each costs incurred by patients and guardians (if applicable) as the purpose is to estimate total costs incurred by the households, and therefore, each cost component already included the costs incurred by guardians.

The WHO handbook proposes two methods in estimating indirect cost: output approach and human capital approach. For the former, time spent by patients or guardians for accessing and receiving care will be summed up and monetised thus it is possible to have disaggregation by time (e.g. before and after diagnosis) and by service. 

The latter approach, output approach, uses income loss, which is the difference of reported income after and before TB diagnosis. By nature of this definition, income loss cannot be disaggregated as it represents the reduction of income during the whole duration of treatment as a single value. 

The WHO handbook advises countries to choose either of the methods considering the specific country context. 

In our study, although we used both approaches for initial analysis, we decided to employ output approach for our main analysis through a consultation process with government representatives, national and international experts. The main reason was that the loss of income apparently captured social consequences of TB-affected household (e.g. job loss) much more than the simple sum of “lost hours”. 

Therefore the result presented in this paper uses output approach with a single value that represents the loss of income for the entire episode of TB. 

11 Table 7. Put ‘Ref’ or ‘1’ in OR column for reference variables.

 We added Ref for each reference variable in the table.

12 Table 7. Authors analyzed determinants of catastrophic costs for mixed group (DS and DR-TB). The costs in DR-TB group were much higher than the costs incurred by DS-TB group. Why did authors mix the groups in the analyses instead of separate the groups? Since the number of DR-TB subjects are very few, it may be enough to analyze it in descriptive way rather than using regression? Please consult statistician regarding this issue.

 Thank you for your suggestion. Even though the limited number of the sample with DR-TB (and relatively large p-value), it has still value to show the high OR to face catastrophic costs among DR-TB patients compared to DS-TB patients, and therefore we would keep the analysis as it is without excluding DR-TB patients.

13 Lines 368: “In Lao PDR, the NTP implements a cash transfer programme for DR-TB patients that provides US$ 5 per day to support expenses for food and transportation.” Was the cash transfer included in costs calculation? Please also confirm it in method section including how the transfer is delivered (condition, unconditional, etc.).

 Thank you for your comment. When we did interviews to healthcare workers in PMDT, we realized that some portion of the transferred cash (USD 5 per day) was taken by the health facilities to provide daily meals to DR-TB patients in TB ward, and only the rest were given to the patients (or their household members). Therefore it was difficult to distinguish the amount for food and others. Another reason is, as in line 270, that only 10% of DR-TB patients reported that they were receiving TB-specific cash transfer or supports though all of them supposed to be eligible to receive it. To avoid under-estimation of the costs (or over-estimation of impact of the cash transfer system), we did not deduct the amount of cash transfer from the estimated costs.

Reviewer 2

1 Line 80: Please describe any prior costing work done in Laos regarding healthcare in general and for TB in particular, if any, besides using the WHO patient cost survey tool.

 No cost surveys or studies related to TB were conducted from patient perspective prior to this survey. 

Other OOP studies exist particularly for MCH e.g.:

- Health care expenditure for hospital-based delivery care in Lao PDR” BMC Research Notes volume 5, Article number: 30 (2012) 

- The Impact of Out-of-Pocket Expenditures on Families and Barriers to Use of Maternal and Child Health Services in the Lao People’s Democratic Republic: Evidence from the Lao Expenditure and Consumption Survey 2007–2008 RETA–6515 Country Brief. Manila: Asian Development Bank. 

2 Line 132: Please add text to justify the selection of 50% catastrophic cost incurrence in your sample size estimate.

 We assumed incidence of 50% catastrophic costs in Lao PDR from the previous TB-PCS in the WPR (PHL: 35%, VNM: 63%, MNG: 70%), and the unweighted average was 56%. We added this sentence in the method section. In addition, an estimated prevalence of 50% will provide the most conservative sample size. 

3 Line 133: Please describe the reason for separating the initial nationally representative sample and the additional sample, and particularly please describe whether the 30 MDR-TB patients from the additional sampling were included in the DS-TB vs MDR-TB comparisons as well as the regression analysis. If they were excluded, please explain why.

 The reasons for separating two samples are:

1. Due to different sampling method. The national samples were enrolled with cluster randomized sampling to ensure the national representativeness. On the other hand, the additional samples of DR-TB and TB-HIV co-infected patients were purposively enrolled from the facilities where provide treatment for DR-TB and ART.

2. Different sampling period. The data collection for the national samples was conducted in December 2018 to January 2019 while that for the additional samples were separately carried out in May and June 2019.

4 Lines 139-143: Please describe the standard treatment length of MDR-TB in Laos and among the MDR-TB cases included on the survey. Especially note if any patients were on shortened MDR-TB regimen. Please do the same for extra-pulmonary TB and provide the average/median treatment duration for the 52 EP-TB patients in the study.

 Although both conventional and shorter regimens were available at the time of this survey, all DR-TB patients in our study were being treated with shorter regimen. We added the comparison of treatment duration in table 2.

5 Line 158: It is not clear how conducting each survey only once minimizes recall bias. Instead, a longitudinal survey may have been the more appropriate way to minimize recall bias. As such, perhaps remove the phrase “To minimize recall bias.”

 Thank you for this suggestion. We removed the phrase.

6 Lines 159-160: Presumably you asked about income prior to the current episode of TB (as per WHO patient cost tool) along with current income, so it is probably best to list that here as well.

 Thank you for your suggestion. We added a sentence “We estimated income loss in TB patients’ households using the income prior to the current TB episode and that at the time of interview (output approach).”

7 Lines 161-162: Please describe how you treated caregiver time loss and costs, i.e., were these included in the cost calculation.

 Since the estimation of income loss in patient’s households was carried out using output approach (based on reported household income), caregiver (or household member’s) time loss was not translated into monetary value and not used for estimating income loss.

However, cost incurred by caregivers and/or household members (such as transportation, food, accommodation for accompanying with TB patients to visit health facilities) were included in the cost calculation.

8 Lines 161-162: Please mention and explain further how time loss was translated into monetary value (i.e., human capital approach, minimum hourly wage...).

 Please see the response above (No.7).

9 Lines 162-164: Since pre-treatment costs were not asked of participants in the CP (62.8% of participants), clearly describe the assumptions made in the extrapolation of pre-treatment costs for the whole sample.

 Thank you for your suggestion. We added a sentence:

“For example, to estimate pre-treatment costs and costs during TB intensive phase for patients who were in continuation phase at the time of interview, the median costs of pre-treatment costs and intensive phase were taken from the patients who were in intensive phase at the time of interview. In this calculation for extrapolating costs, costs from DS-TB and DR-TB patients, and patients with and without hospitalizations were considered separately.”

10 Lines 187-188: Please describe how you treated missing values in the responses (besides the imputed household income).

 First, when the tablet-based questionnaire was developed and adapted to Lao context, we set restrictions not to miss essential data during data collection. Then, the data was intensively checked by Yamanaka T for data cleaning and validation. During the validation process, identified missing data and outliers were returned to survey implementers and data collectors and corrected.

11 Lines 187-188: Please describe how you treated hospitalization costs and whether you were able to track any hospitalization costs after the survey was performed (but before the end of treatment).

 Following the recommended method by the WHO (and in the handbook for TB patient cost survey), the costs for hospitalization was simply extrapolated until the end of the current treatment phase

12 Lines 187-188: If household assets were used for imputation of HH income, please furnish supplemental information on the types, frequencies and regression results for the imputed HH income as well as the proportion of participants for whom imputation had to be employed

 We added the information as supplementary material 3

13 Lines 187-188 & 194-198: Did your regression analysis for imputation of HH income and for evaluation of association of patient covariates with catastrophic cost incurrence account for clustering effects and intra cluster correlation from your sampling strategy? If it did, please describe in further detail in the paper the regression methods used. If not, please re-fit your model using appropriate specifications including a description of how you arrived at the final model specifications or explain why you chose not to account for ICC

 Thank you for your suggestion. We took the adjustment into account in the analysis. Considering changes in ORs with the adjustment, we changed the table 7 with results with the adjustment for sampling method. Also, after adjusting sampling method, the overall % of catastrophic costs was changed to 62.6% (0.1% increase from the previous result). We also changed figure 2 accordingly.

14 Line 195-196: Please describe and provide examples how including the additional sample of MDR-TB and TB-HIV participants would have affected the regression analysis.

 If the national samples and additional samples were enrolled in a same sampling method, and all the samples were included in the logistic regression analysis, statistical significance in drug-resistance status and HIV status in % of facing catastrophic costs could have been shown in table 7. However, due to the different sampling framework, we excluded the additional samples from statistical comparisons and risk factor analysis.

15 Line 317: Please describe wealth quintile in this section rather than the results and elaborate on what "based on household income" means. If it is simply quintiles of Household income then please explain and label it so, since wealth can include other tangible and intangible factors.

 Thank you for your suggestion. We removed the phrase.

16 Lines 216-217: It is very surprising to see the proportion of participants without health insurance reported as 75.9% when the national statistics claim that insurance coverage is 60%, 70% or even 79.3% in 2019. Please furnish the Insurance coverage rate among TB patients notified by the Lao NTP to understand if there is any bias in the study sample.

 Thank you for your comment. 

There is no bias in the coverage since the NHI covers all the population regardless of TB or not. The result of this survey revealed a less recognition of NHI coverage in TB patients although the NHI was implemented in 2016 (2 years back from the time of this survey).

17 Lines ibid: If the average national insurance coverage among TB patients is really as low as found in the patient first survey, please include this fact in the study setting section.

 Please see the response above (No. 16).

18 Line 228: The proportion of TB patients in the results treated at public health and district facilities states 61% for the national sample, while the table shows 71%. Please review all of your numbers again and ensure there is internal consistency.

 We apologize this typo, and it is corrected. 

19 Lines 228-229: Please provide a post-hoc analysis (as supplemental information) of the CC incurrence in the subpopulation of the 29.9% of participants treated at the provincial/national facilities. For this post-hoc analysis, test whether the subpopulations are significantly different from each other.

 Thank you for your suggestion. Supplemental material 4 is added for this supplementary analysis and no significant difference in incurrence of catastrophic costs was found (p-value: 0.197).

20 Lines 239-240: Given that 97.7% of the national sample was treated under SAT, please provide information on treatment monitoring standards under national TB treatment guidelines in the methods. Please discuss and clarify the root cause behind the high rate of income loss under SAT

 Although DOT by a health worker is recommended as a treatment monitoring standard in National TB guidelines to ensure adherence, in practice TB patients are often living far and manage to collect TB drugs intermittently from nearest facilities (frequency of visits: once a week to once a month), and take the TB medications with village volunteer or family member support only.

The reasons of income loss are:

1. Even with self-administered therapy, patients (and/or their household members) have to visit facilities for the drug pick-up

2. Patients can lose their jobs due to being too unwell to work or their household members can lose their jobs (and/or time to work) to support TB patients in their households. 

21 Line 580: Please describe the discrepancies in sample sizes between the total participant figure (n=725) in the national sample and respondents for time loss due to TB (n=292).

 We are sorry for this typo, and we corrected this error.

22 Line 337: please reference any sources that have formally assessed and linked dissavings and asset sale to prolonged negative impact on their lives.

 We added a reference.

23 Line 391: Please rephrase the word minimal perhaps as 16% of one's income is not necessarily minimal, especially for low income households.

 Thank you for your suggestion. We rephrased it to “the direct medical costs had relatively smaller impact on total TB patient costs in this survey (15.8%)”

24 Line 392: "Ensuring the future sustainability of free and high-quality TB services..." is perhaps more of a basic essential for a TB program rather than a strong discussion point. Sustained free and high-quality TB services is the bare minimum expectation and without it, there should be no End TB Strategy. As this paragraph talks about diagnostic delay and direct medical costs, perhaps identify interventions from the literature that have shown to reduce these barriers to care. One such place could be line 389, which should include an example or at minimum a few references.

 Thank you for your suggestion. We added statements as;

“Implementing active case finding (ACF) may result in a shorter delay in TB diagnosis compared to facility based passive case finding (PCF) as well as reduction in costs incurred by TB patients [44, 45]. A study in Cambodia comparing TB patient costs between ACF and PCF revealed that costs before TB diagnosis was significantly lower among patients detected with ACF compared to those with PCF whereas no difference was found in costs during TB treatment [39]. Furthermore, increasing awareness of the NHI would be also important to have early diagnosis of TB. Our study showed a considerably low recognition of NHI coverage among TB patients (75.9% reported no insurance) even though the NHI was implemented in 2016 (2 years back from the time of this study). This low recognition of NHI could be a barrier to use healthcare services in public facilities after having TB symptoms.”

25 Line 398: The discussion of informally employed participants seems insufficient and a disproportional amount of text is dedicated to formally employed persons who comprise less than 10% of the sample. Hence, please expand the discussion on informally employed participants. Specifically, in the absence of any unemployment protection schemes, which in LMIC often only apply after having paid into the scheme through formal employment, the question for informally employed persons is how fast they were able to recover and pick up their employment again. Informal employment usually has lower barriers to entry into the job market (e.g., rag picking, selling lottery tickets, street-side parking attendants, etc), so perhaps the discussion can allude to the current lack of evidence on post-treatment socioeconomic recovery.

 Thank you for your suggestion. We added discussions as:

“Implementing an additional cash transfer programme would be one option to address this issue. In Vietnam, after conducting TB patient cost survey, the NTP implemented an innovative financial support scheme, so called “Patients Support Foundation to End Tuberculosis (PASTB)” for TB patients in poverty using a short message service (SMS) [46]. In this campaign, every SMS will transfer US$0.8 to the fund to provide financial support for TB patients [46]. In addition to the supports during TB treatment, the post-treatment socio-economic recovery (e.g. re-employment after TB treatment) is also important to minimize long-term or permanent financial impacts due to TB. However, no studies assessing long-term economic shocks in TB-affected households after completion of TB treatment (e.g. permanent job loss, continuous social exclusion) are published yet except for an on-going study in African countries [47]. More evidences around post-treatment economic impacts due to TB are required especially in Asian contexts.”

26 Lines 175-177 & Lines 425-428: The reasoning for not conducting statistical analysis between DS-TB patients (n=717) and TB/HIV patients (n=123) or even MDR-TB patients (n=30) when you performed statistical comparisons between DS-TB and MDR-TB (n=8) within the national sample is not clear. The reasoning for not performing statistical comparisons between the methods and discussion also seem discordant. As such, please rephrase to provide a more plausible reason for the lack of statistical comparison with the additional sample.

 The reasons for not conducting statistical analysis with two samples are:

1. Due to different sampling method. The national samples were enrolled with cluster randomized sampling to ensure the national representativeness. On the other hand, the additional samples of DR-TB and TB-HIV co-infected patients were purposively enrolled from the facilities where provide treatment for DR-TB and ART.

2. Different sampling period. The data collection for the national samples was conducted in December 2018 to January 2019 while that for the additional samples were separately carried out in May and June 2019.

We rephrase the limitation as:

“Second, 123 additional patients (22 DR-TB patients and 101 TB-HIV coinfected patients) were enrolled with purposive sampling at different time (in May-June 2019) separately from the 725 nationally representative sample (in December 2018-January 2019). Thus, statistical tests comparing them with the national sample were not carried out for this reason.”

27 Line 427: Please remove the word "due".

 We removed the word.

28 Line 432: Please change "suspected TB cases" to persons with presumptive/suspected TB to avoid the use of stigmatizing language.

 We changed it to “all the individuals with suspected TB”.

29 Line 434: "Therefore, we assumed that the impact of this exclusion criteria would be minimal." - to be able to make this assumption, it would be necessary to have information on the proportion of missed cases and proportion treated in the private sector (onion model). Please furnish this information or consider rephrasing to not state a potentially false assumption.

 Thank you for your suggestion. We removed the sentence “Therefore, we assumed that the impact of this exclusion criteria would be minimal.”

30 Lines 435-436: this is a very good and critical point. Please provide a map/table in the supplemental information of sample sizes by geography to identify any potential source for bias

 Thank you for your suggestion. We included a map and a table showing the number of clusters in each province as supplementary material 2.

---

## [Decision Letter · Decision Letter 1]

22 Oct 2020

PONE-D-20-27870R1

First national tuberculosis patient cost survey in Lao People’s Democratic Republic: Assessment of the financial burden faced by TB-affected households and the comparisons by drug-resistance and HIV status

PLOS ONE

Dear Dr. Yamanaka,

Thank you for submitting your manuscript to PLOS ONE. After careful consideration, we feel that it has merit but does not fully meet PLOS ONE’s publication criteria as it currently stands. Therefore, we invite you to submit a revised version of the manuscript that addresses the points raised during the review process.

Please address the outstanding comments made by Reviewer 1, which include: clarifying the additional sampling, discussing the limitations of the DR-TB sampling and sample size, interpreting the findings of the HIV-TB co-infected population in the cohort, and rectifying Figure issues.

We look forward to receiving your revised manuscript.

Kind regards,

Tom E. Wingfield

Academic Editor

PLOS ONE

Reviewers' comments:

Reviewer's Responses to Questions

**Comments to the Author**

1. If the authors have adequately addressed your comments raised in a previous round of review and you feel that this manuscript is now acceptable for publication, you may indicate that here to bypass the “Comments to the Author” section, enter your conflict of interest statement in the “Confidential to Editor” section, and submit your "Accept" recommendation.

Reviewer #1: (No Response)

2. Is the manuscript technically sound, and do the data support the conclusions?

Reviewer #1: Partly

3. Has the statistical analysis been performed appropriately and rigorously? 

Reviewer #1: Yes

4. Have the authors made all data underlying the findings in their manuscript fully available?

Reviewer #1: Yes

5. Is the manuscript presented in an intelligible fashion and written in standard English?

Reviewer #1: Yes

6. Review Comments to the Author

Reviewer #1: Authors have responded almost all of previous comments. However, some points remain for improvement.

1. The reasons of using additional sample remains inconsistent and confusing. In lines 111-113, authors wrote that additional respondents were aimed “to assess the difference in the financial burden of TB comparing drug-resistant and drug-susceptible TB patients and for patients with and without TB-HIV co-infection”. However, in lines 156-158, authors stated “Due to different sampling methods for nationally representative sample and additional sample of DR-TB and TB-HIV coinfected patients, statistical tests were performed only in nationally representative sample.” Did authors mean the “assessing the difference” as only in a descriptive way?

2. Authors added a group of TB-HIV, but lacked of discussion on this group in discussion section. It is better to discuss the findings in this group if authors thought that adding this group to the sample is really needed.

3. The number of DR-TB patients in national sample were very few and were considerably not enough for statistical comparison. Better if authors mention this shortage in limitation.

4. Table 3 : Some cells show hours lost in negative values, e.g. -212.2-515. It seems odd, and better to show the values in median (range) instead of mean (95% CI).

5. Figure 3 is missing.

6. Avoid writing numbers at the beginning of a sentence.

7. PLOS authors have the option to publish the peer review history of their article (what does this mean?). If published, this will include your full peer review and any attached files.

Reviewer #1: No

---

## [Author Response · Author response to Decision Letter 1]

22 Oct 2020

Reviewer 1

1 The reasons of using additional sample remains inconsistent and confusing. In lines 111-113, authors wrote that additional respondents were aimed “to assess the difference in the financial burden of TB comparing drug-resistant and drug-susceptible TB patients and for patients with and without TB-HIV co-infection”. However, in lines 156-158, authors stated “Due to different sampling methods for nationally representative sample and additional sample of DR-TB and TB-HIV coinfected patients, statistical tests were performed only in nationally representative sample.” Did authors mean the “assessing the difference” as only in a descriptive way?

 Thank you for your comment. Yes, we aimed at assessing the costs among DR-TB and TB-HIV co-infected patients only in a descriptive way due to the different sampling method and also limited number of samples for DR-TB (even though we enrolled total samples at the time of the survey). 

2 Authors added a group of TB-HIV, but lacked of discussion on this group in discussion section. It is better to discuss the findings in this group if authors thought that adding this group to the sample is really needed.

 Thank you for your suggestion. We added a discussion for TB-HIV as:

Integrated services for TB and HIV was provided only at 11 of 25 central and provincial hospitals, and therefore patients with TB-HIV coinfection had to travel to those hospitals that are usually located far from their residences compared to public health centers or district hospitals, or had to have separate facility visits for TB and HIV treatments [26-28]. Enhancing and decentralizing integrated services for TB and HIV would be necessary to mitigate the financial burden in TB-HIV coinfected patients.

3 The number of DR-TB patients in national sample were very few and were considerably not enough for statistical comparison. Better if authors mention this shortage in limitation.

 Thank you for the comment. We added following sentences for limitation 2 as:

Furthermore, although we conducted statistical tests comparing various factors between DS-TB and DR-TB patients in the national sample, the number of DR-TB patients in the national sample was very few (N=8) and not enough especially for risk factor analysis for facing catastrophic costs.

4 Table 3 : Some cells show hours lost in negative values, e.g. -212.2-515. It seems odd, and better to show the values in median (range) instead of mean (95% CI).

 Thank you for your suggestion. We revised table 3 (from mean to median)

5 Figure 3 is missing.

 We attached all the figures in the submission

6 Avoid writing numbers at the beginning of a sentence.

 We revised those sentences (mainly in result section).

---

## [Editor Report · Decision Letter 2]

22 Oct 2020

First national tuberculosis patient cost survey in Lao People’s Democratic Republic: Assessment of the financial burden faced by TB-affected households and the comparisons by drug-resistance and HIV status

PONE-D-20-27870R2

Dear Dr. Yamanaka,

We’re pleased to inform you that your manuscript has been judged scientifically suitable for publication and will be formally accepted for publication once it meets all outstanding technical requirements.

Kind regards,

Tom E. Wingfield

Academic Editor

PLOS ONE
---

## [Editor Report · Acceptance letter]

29 Oct 2020

PONE-D-20-27870R2 

First national tuberculosis patient cost survey in Lao People’s Democratic Republic: Assessment of the financial burden faced by TB-affected households and the comparisons by drug-resistance and HIV status 

Dear Dr. Yamanaka:

I'm pleased to inform you that your manuscript has been deemed suitable for publication in PLOS ONE. Congratulations! Your manuscript is now with our production department. 

Kind regards, 

on behalf of

Dr. Tom E. Wingfield 

Academic Editor

PLOS ONE